# Selective control of synaptically-connected circuit elements by all-optical synapses

Mansi Prakash [1], Jeremy Murphy[2], Robyn St Laurent [3,8], Nina Friedman[2], Emmanuel L. Crespo[4], Andreas Bjorefeldt[1,2,9], Akash Pal [5,10], Yuvraj Bhagat[1], Julie A. Kauer [3,11], Nathan C. Shaner[6], Diane Lipscombe [7], Christopher I. Moore [7✉] & Ute Hochgeschwender [1,4,5✉]

Understanding percepts, engrams and actions requires methods for selectively modulating synaptic communication between specific subsets of interconnected cells. Here, we develop an approach to control synaptically connected elements using bioluminescent light: Luciferase-generated light, originating from a presynaptic axon terminal, modulates an opsin in its postsynaptic target. Vesicular-localized luciferase is released into the synaptic cleft in response to presynaptic activity, creating a real-time Optical Synapse. Light production is under experimenter-control by introduction of the small molecule luciferin. Signal transmission across this optical synapse is temporally defined by the presence of both the luciferin and presynaptic activity. We validate synaptic Interluminescence by multi-electrode recording in cultured neurons and in mice in vivo. Interluminescence represents a powerful approach to achieve synapse-specific and activity-dependent circuit control in vivo.

[1] College of Medicine, Central Michigan University, Mt Pleasant, MI, USA. [2] Department of Neuroscience, Brown University, Providence, RI, USA. [3] Department of Molecular Pharmacology, Physiology, and Biotechnology, Brown University, Providence, RI, USA. [4] Biochemistry, Cellular and Molecular Biology Program, Central Michigan University, Mt Pleasant, MI, USA. [5] Neuroscience Program, Central Michigan University, Mt Pleasant, MI, USA. [6] Department of Neuroscience, University of California San Diego, La Jolla, CA, USA. [7] Carney Institute for Brain Science, Brown University, Providence, RI, USA. [8] Present address: The Gladstone Institutes, San Francisco, CA, USA. [9] Present address: Institute of Neuroscience and Physiology, University of Gothenburg, Gothenburg, Sweden. [10] Present address: Department of Biochemistry and Molecular Medicine, School of Medicine, UC Davis, Davis, CA, USA. [11] Present address: Department of Psychiatry and Behavioral Sciences, Stanford University School of Medicine, Stanford, CA, USA. ✉email: christopher_moore@brown.edu; hochg1u@cmich.edu

A wealth of new tools are revolutionizing neuroscience by allowing direct control of specific subpopulations of neurons for brief and sustained time periods (e.g., opto-, chemo- and sonogenetics;[1–3]). The ability to regulate genetically identified neurons in selected brain areas has been a significant benefit to studying neural dynamics and its link to behavior.

However, information processing leading to percepts, memories and/or actions requires multiple nodes acting in sequence within a network, with multiple cell types in multiple areas conducting specific cell-to-cell communication. The ideal tool(s) in the next generation of approaches will allow intersectional circuit dissection—specifying and regulating participants at multiple stations. Further, tools for fully understanding systems underlying behavior will allow them to demonstrate natural activity, enhancing or suppressing, for example, transmission of endogenous patterns. While real-time feedback interventions driven by computer recognition of activity patterns are increasingly being applied to electrical stimulation (e.g., deep-brain stimulation) systems, most tools that have genetic precision and molecular specificity are still regulated *en masse* by rising gradients of sustained chemical drivers or by imposed patterns of optogenetic drive. Tools that permit direct experimental control of the efficacy and form of synaptic transmission between specific partners, will be a key step in providing the next wave of insight into network dynamics and function.

The most common strategy currently in use to modulate specific synaptic connections (Fig. 1a) involves light-activation of opsin-expressing presynaptic neurons using localized fiber optic stimulation near the postsynaptic target neurons (Fig. 1b[4]). A high degree of presynaptic specificity can be achieved by this approach, using retrograde and Cre-dependent expression of optogenetic elements and localizing the light source, yet postsynaptic specificity will depend on the extent of presynaptic contact spread. A conceptually similar approach is achieved using chemogenetic methods to modulate synaptic transmission by targeting axon terminals (Fig. 1c). Chemogenetic neuromodulation can be restricted to a subset of target neurons, leaving other interconnected areas unaffected. For example, designer-receptors-activated-exclusively-by-designer-drugs expressed in long-range projecting neurons in one cortical layer can be activated by ligand application to a confined area in another cortical layer[5]. These methods are limited by studying cell-cell communication between populations that are anatomically separate because 1 mm³ of light (optogenetic) or exogenous drug (chemogenetic) will likely act on multiple cells. Further, most neural computations take place between highly interspersed cells in tightly packed spaces.

Here, we describe the Optical Synapse, an approach to control synaptic connectivity that utilizes presynaptically originating bioluminescence to activate optogenetic actuators expressed at postsynaptic sites (Fig. 1d). We have shown that bioluminescent optogenetics can operate within a cell, with a luciferase tethered to an opsin (a Luminopsin)[6–9]. Photon release from the luciferin-luciferase interaction activates the associated opsin thereby achieving optogenetic modulation. Depending on the biophysical properties of the opsin, excitatory or inhibitory, bioluminescence may depolarize or hyperpolarize the neuron.

In our *Interluminescent* Optical Synapse, we used bioluminescent optogenetics to achieve synapse-specific and activity-dependent circuit control, by expressing the luciferase presynaptically and its partner opsin postsynaptically. When presynaptic luciferase and postsynaptic opsin are present at the same synapse, bioluminescent optogenetic modulation is triggered when luciferin is provided, a requirement that allows experimental control of intersectional communication. The spatial requirements of Interluminescence in the current application restricts it to synapses that express the luciferase in presynaptic vesicles and opsins postsynaptically. The release of luciferase from the presynaptic terminal depends on presynaptic depolarization, similar to synaptic transmission via neurotransmitters or neuropeptides. Here we describe examples

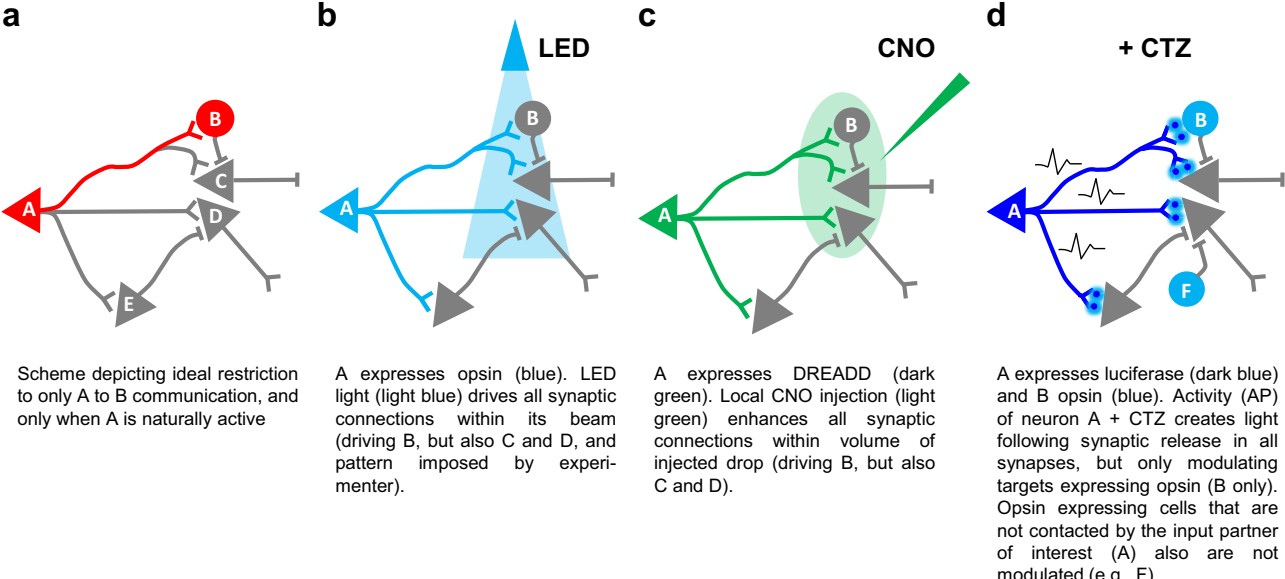

**a**

Scheme depicting ideal restriction to only A to B communication, and only when A is naturally active

**b** LED

A expresses opsin (blue). LED light (light blue) drives all synaptic connections within its beam (driving B, but also C and D, and pattern imposed by experimenter).

**c** CNO

A expresses DREADD (dark green). Local CNO injection (light green) enhances all synaptic connections within volume of injected drop (driving B, but also C and D).

**d** + CTZ

A expresses luciferase (dark blue) and B opsin (blue). Activity (AP) of neuron A + CTZ creates light following synaptic release in all synapses, but only modulating targets expressing opsin (B only). Opsin expressing cells that are not contacted by the input partner of interest (A) also are not modulated (e.g., F).

**Fig. 1 Transsynaptic modulation. a** Cell A connects with cells B, C, D and E; however, only cell A's communication with cell B should be modulated, either amplified or dampened. **b** If cell A expresses optogenetic actuators, restriction of a light beam to the area of intended synaptic transmission can minimize unwanted activation (cell E will not be activated), but the likelihood of still activating unwanted synapses (cells C and D) is high. **c** The same applies when expressing a chemogenetic actuator (DREADD, designer receptor exclusively activated by designer drugs) in cell A and restricting application of the ligand CNO (*Clozapine*-N-Oxide) to an anatomical area as small as possible. **d** True cell-to-cell synaptic communication can be achieved by expressing a luciferase in cell A and an opsin in cell B. Activity of neuron A and application of the luciferin CTZ (Coelenterazine) results in light emission at all synapses of A, but only the opsin-expressing cell B will be modulated. At the same time, opsin-expressing cells not synaptically contacted by the luciferase-expressing cell A will not be modulated.

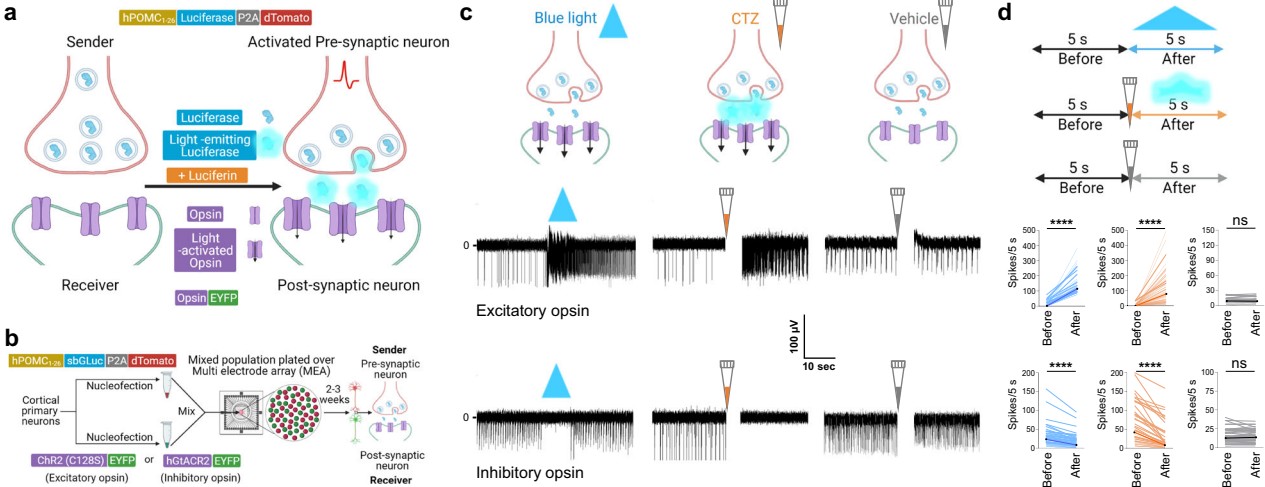

**Fig. 2 Modulation of postsynaptic neural activity by Interluminescence in mixed cultures of pre- and postsynaptic neurons. a** Schematic: Interluminescence via an Optical Synapse. Luciferases (blue colored enzyme inside the gray circles) are released from presynaptic vesicles and, in the presence of the luciferin, emit light (bioluminescence: light bluish glow) that activates postsynaptic opsins (magenta; downward black arrows indicating ion movement through open channels). **b** Schematic of experimental design and constructs used for separate nucleofections of cortical neurons with either hPOMC1-26-sbGLuc-P2A-dTomato or one of the opsins, ChR2(C128S)-EYFP (excitatory) and hGtACR2-EYFP (inhibitory); neurons were then mixed (red and green spheres), plated on MEAs and maintained for 2–3 weeks until recording. **c** Illustrations (upper panels) and corresponding representative traces from individual electrodes of MEAs (middle and lower panels) showing response of postsynaptic neurons to blue light (blue solid triangle), CTZ-induced bioluminescence (orange pipette tip) and vehicle (gray pipette tip) expressing the excitatory opsin ChR2(C128S) (middle panels) and the inhibitory opsin hGtACR2 (lower panels). **d** Schematic showing the time windows for analyzing the number of spikes for each treatment (upper panel) and ladder plots from multiple experiments (middle panels for excitatory and lower panels for inhibitory opsin-expressing postsynaptic neurons as depicted for individual traces in (**c**). ChR2(C128S), blue light, $n = 62$, $p < 0.0001$, CTZ, $n = 62$, $p < 0.0001$, vehicle, $n = 24$, $p = 0.6498$; hGtACR2, blue light, $n = 49$, $p < 0.0001$, CTZ, $n = 49$, $p < 0.0001$, vehicle, $n = 49$, $p = 0.5594$; Wilcoxon matched-pairs signed rank test. The artifacts due to addition of reagents in MEAs are overlaid by a vertical white bar in all MEA recording traces (the white gap right after addition of either CTZ or vehicle). ns, not significant; ****$p < 0.0001$.

of the transmission of bioluminescence signals across synapses in culture and in vivo, and through a series of independent tests show that the postsynaptic output is mediated by optical coupling, and independent of classic neurotransmitter-mediated synaptic transmission. This intersectional technology can provide a novel class of cell-pairing and activity-specific control for testing the mechanisms underlying behavior.

## Results

**Interluminescence via an Optical Synapse.** We used cultured cortical neurons to first establish if a luciferase genetically targeted to presynaptic vesicles could be released into the synaptic cleft and generate sufficient photon density to activate opsins genetically targeted to a postsynaptic cell. We term this form of bioluminescence-mediated synaptic transmission Interluminescence (Fig. 2a). Critically, we needed to show that Interluminescence: (i) generates a measurable postsynaptic response; (ii) mediates different postsynaptic responses depending on the type of opsin, with cation permeable opsins triggering postsynaptic depolarization and excitation, and anion permeable opsins hyperpolarization and inhibition; (iii) occurs when luciferase is co-released with endogenous transmitters and peptides, but only in the presence of the luciferin; (iv) occurs independent of classic neurotransmission; and, (v) can co-exist with classic neurotransmission.

**Interluminescence modulates spontaneous neural activity in culture.** We targeted the blue light-emitting luciferase sbGLuc, a bright *Gaussia* luciferase variant[10], to vesicles in cortical neurons using the vesicle targeting sequence of the human pro-opiomelanocortin pro-peptide (hPOMC1-26)[11,12]. The targeting construct also contained the reporter gene dTomato attached to sbGLuc via a P2A cleavage sequence (Fig. 2a). In addition to

being a bright photon source, sbGLuc is favorable because it is also stable at the lower pH levels of synaptic vesicles[13,14].

We selected high light sensitivity opsins for our initial experiments. We employed the excitatory step-function opsin ChR2(C128S)[15] and the inhibitory anion channel hGtACR2[16] as both exhibit high sensitivity to blue light relative to other opsins. In our initial studies, we separately nucleofected cortical neurons with hPOMC1-26-sbGLuc, ChR2(C128S), or hGtACR2 to ensure that luciferase and opsin were expressed in different cell populations (Fig. 2b). We then mixed the two populations of cells, one luciferase expressing and one opsin expressing, and plated the mixture on multielectrode arrays (MEAs)[17]. We used externally presented blue light to activate opsins directly and showed that this increased (Fig. 2c middle panel; ChR2(C128S)) or decreased (Fig. 2c, lower panel; hGtACR2) the activity of the culture, as expected based on the type of opsin expressed. We then added the luciferin for *Gaussia* luciferases, Coelenterazine (CTZ), and observed increased (ChR2(C128S)) or decreased (hGtACR2) spontaneous activity consistent with the expressed opsin (Fig. 2c). By contrast, the addition of the vehicle alone had no consistent impact on ongoing neural activity.

We used direct LED stimulation of postsynaptic opsins as an internal control and compared the responses of cultures to LED (blue light), bioluminescence (CTZ), and control (vehicle) in three independent experiments (summarized in the ladder plots in Fig. 2d; Supplementary Data 1). Spontaneous activity levels varied across MEAs as expected (see also Supplementary Fig. 1). We measured spike rate (number of spikes/s) before and after treatment to compare each manipulation across different electrodes and MEAs. We observed a significant difference in spontaneous activity in cultures before and after stimulation by blue light LED and CTZ but no consistent differences in neural activity before and after vehicle (ChR2(C128S), blue light, $n = 62$,

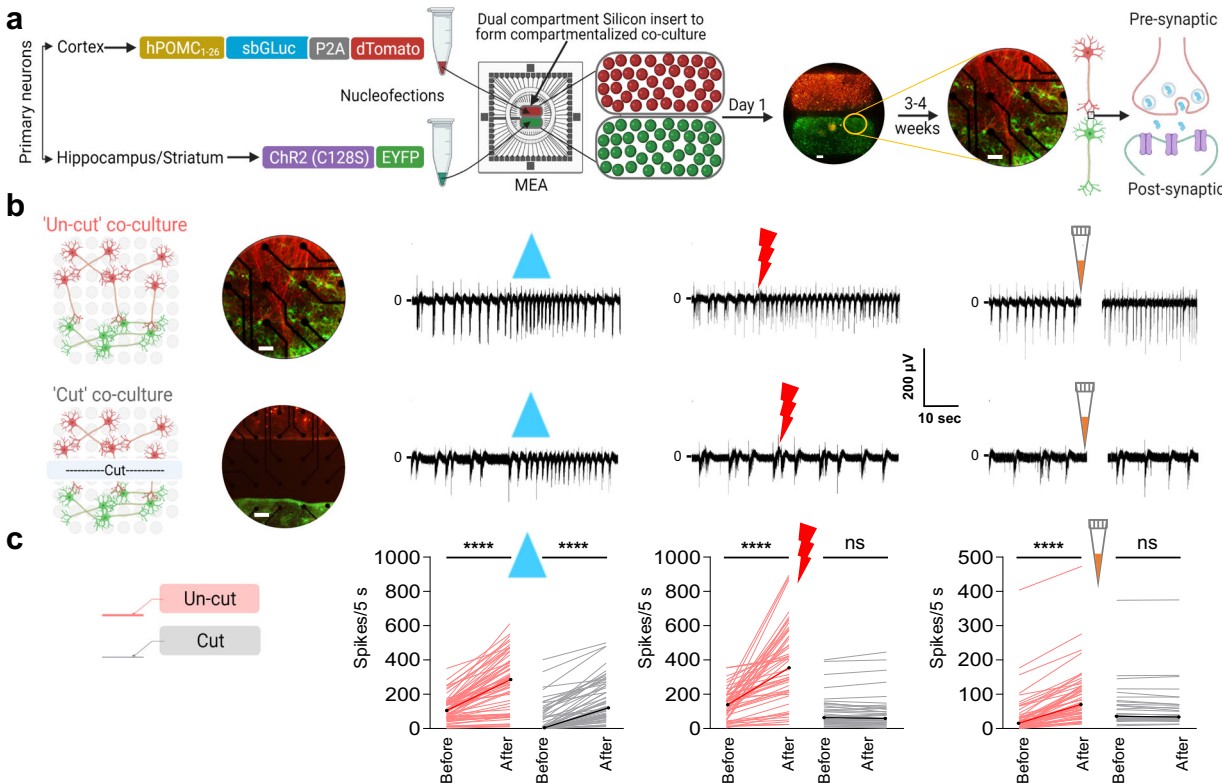

**Fig. 3 Communication via Interluminescence in co-cultured populations depends on intact synaptic connections. a** Schematic of experimental design and constructs used for separate nucleofections of cortical and hippocampal or striatal primary neurons. Cortical neurons were nucleofected with the luciferase construct (hPOMC1-26-sbGLuc-P2A-dTomato) and plated in the upper compartment of a 2-chamber silicon divider (group of red spheres covering the upper half of MEA electrodes), and their natural synaptic targets, hippocampal or striatal neurons, were nucleofected with the excitatory opsin construct (ChR2(C128S)-EYFP) and plated in the lower compartment (group of green spheres covering the lower half of MEA electrodes). The next day neurons had attached and the divider was removed (fluorescent image; ×5 magnification; scale bar: 216 μm) showing the expression of luciferase and opsin in respective neuronal populations. Cultures matured over the next 3 weeks, with processes from cortical neurons growing deep into the hippocampal or striatal areas (fluorescent image; ×20 magnification; scale bar: 54 μm) showing the processes from the cortical population (dTomato) contacting the hippocampal neurons (EYFP). **b** Illustrations showing the layout of electrodes (light gray circles) in 1-well MEAs with a co-culture (left panel) when the inter-population connections are intact (un-cut: upper left) or severed (cut: lower left) by making a cut between the two populations (fluorescent images; 20x magnification; scale bar: 54 μm). Recordings from one electrode within the postsynaptic population (right panels) when treated with blue light (blue solid triangle), presynaptic electrical stimulation (red bolt), and CTZ (orange pipette tip) with connecting processes intact (right upper) versus cut (right lower). **c** Schematic showing the color code (left panel) used for the ladder plots for both un-cut (peach) and cut (gray) co-cultures in the right panel. Ladder plots (right panel) showing change in number of spikes of postsynaptic neurons 5 s before and after each treatment (blue light; electrical stimulation; CTZ) for both the un-cut (peach) and cut (gray) conditions (blue light, un-cut n = 63, p < 0.0001, cut n = 58, p < 0.0001; electrical stimulation, un-cut n = 54, p < 0.0001, cut n = 32, p = 0.0965; CTZ, un-cut n = 51, p < 0.0001, cut n = 38, p = 0.7388; Wilcoxon matched-pairs signed rank test). The artifacts due to addition of reagents in MEAs are overlaid by a vertical white bar in all MEA recording traces (the white gap right after addition of CTZ). ns, not significant; ****p < 0.0001.

p < 0.0001, CTZ, n = 62, p < 0.0001, vehicle, n = 24, p = 0.6498; hGtACR2, blue light, n = 49, p < 0.0001, CTZ, n = 49, p < 0.0001, vehicle, n = 49, p = 0.5594; Wilcoxon matched-pairs signed rank test). Our data also show that the activity of opsin-expressing postsynaptic neurons could be augmented, or inhibited depending on the nature of the postsynaptic opsin, following presynaptic activation in the presence of CTZ.

**Interluminescence requires connections between pre- and postsynaptic neurons.** To assess the properties of CTZ-dependent responses in more detail, and directly test if synaptic connectivity is required, we employed a co-culture system in which presynaptic and postsynaptic populations are seeded in separate compartments and then allowed to form synapses across a separating gap (Fig. 3a). We nucleofected presynaptic cortical neurons with luciferase, and postsynaptic hippocampal or striatal target neurons with ChR2(C128S).

We observed an increase in spontaneous activity and in spiking synchrony within the MEAs as the cultures matured (Supplementary Fig. 2). The physical separation of pre- and postsynaptic neurons allowed us to identify the active population in different treatment conditions (Supplementary Fig. 3). Direct electrical stimulation of presynaptic neurons increased spiking in ChR2(C128S) expressing postsynaptic neurons presumably through classic synaptic neurotransmission (Supplementary Fig. 3c). Blue light LED stimulation (which directly activates the postsynaptic Opsin) or CTZ addition (which depends on ongoing spontaneous presynaptic activity) by contrast, increased spiking in ChR2(C128S) expressing postsynaptic neurons but without a change in the activity of cortical presynaptic neurons (Supplementary Fig. 3d, e). We also showed that only MEA electrodes that responded to direct optogenetic stimulation (LED) responded to CTZ application; an independent demonstration of the specificity of CTZ-dependent responses.

We next tested whether CTZ-dependent changes in post-synaptic neural activity were mediated by synaptic events triggered by presynaptic depolarization. We compared responses before and after severing the connections between the pre- and postsynaptic populations (Fig. 3b, upper versus lower panels). As expected, LED stimulation, which activates postsynaptic opsins directly, induced increased postsynaptic spiking equally in un-cut and cut co-cultures. In contrast, direct electrical stimulation of presynaptic neurons and CTZ-induced bioluminescence failed to alter the excitability of postsynaptic neurons in cut co-cultures. This result is consistent with the hypothesis that CTZ modulation of postsynaptic activity requires synaptic connectivity. The results from several independent experiments of presynaptic (cortical) and postsynaptic (hippocampal, striatal) neural cultures are summarized (Fig. 3c; blue light, un-cut $n = 63$, $p < 0.0001$, cut $n = 58$, $p < 0.0001$; electrical stimulation, un-cut $n = 54$, $p < 0.0001$, cut $n = 32$, $p = 0.0965$; CTZ, un-cut $n = 51$, $p < 0.0001$, cut $n = 38$, $p = 0.7388$; Wilcoxon matched-pairs signed rank test; Supplementary Data 2).

In summary, these data provide strong support for Inter-luminescence as a form of engineered synaptic transmission that achieves cell-cell communication via bioluminescence originating presynaptically activating opsins postsynaptically.

**Interluminescence depends on presynaptic activity and occurs independent of classic synaptic neurotransmission.** We found that addition of a cocktail of transmitter receptor blockers including: NBQX, a selective blocker of non-NMDA mediated synaptic transmission; D-AP5, a NMDA receptor antagonist; Gabazine and CGP55845, antagonists at $GABA_A$ and $GABA_B$ receptors, respectively; and Strychnine, a glycine receptor antagonist, silenced spontaneous activity of the entire neural culture on MEAs for several minutes (Supplementary Fig. 4a, b(i), c; $n = 27$, $P < 0.0001$; Mann–Whitney test; Supplementary Data 3), but activation of postsynaptic opsin-expressing neurons by blue light LED confirmed that neurons responded to direct stimulation (Supplementary Fig. 4b(ii), c; $n = 27$, $P < 0.0001$; Mann–Whitney test; Supplementary Data 3). We employed neurotransmission blockade to test the requirements for bioluminescence-mediated synaptic transmission separate from endogenous transmitter mediated effects (Fig. 4). When CTZ was delivered with, or immediately after, addition of synaptic blockers (SB), postsynaptic activation through bioluminescence was robust, while vehicle addition had no effect (Fig. 4b, c; c(i): SB alone, $n = 27$, c(ii): SB + CTZ, $n = 37$, c(iii): SB + vehicle, $n = 38$; SB alone (after) v/s SB + CTZ (after) $p < 0.0001$; SB + CTZ (after) v/s SB + vehicle (after) $p < 0.0001$; SB alone (after) v/s SB + vehicle (after) $p = 0.7305$; Mann–Whitney test; Supplementary Data 4). However, addition of CTZ more than 20 s after addition of the synaptic blockers had no effect on spiking of opsin-expressing neurons (Fig. 4d(i), e(i): $n = 18$, SB (after) v/s CTZ added later (after) $p = 0.4022$; Mann–Whitney test).

These findings support the following model: In spontaneously active cultures luciferase is present in the synaptic cleft for a certain time period after its release. When CTZ is added immediately after acute synaptic block there is still sufficient luciferase to create photon density great enough to enable opsin activation. In contrast, within 20 s of synaptic block, luciferase levels in the synaptic cleft fall below that which could support opsin activation (silent phase). This interpretation is further supported by the observation that CTZ can induce spiking during the silent phase if it is added immediately following presynaptic electrical stimulation which triggers luciferase release into the synaptic cleft (Fig. 4d, e; e(iii): $n = 10$, electrical stimulation (after) v/s immediately added

CTZ (after) $p < 0.0001$; Mann–Whitney test; Supplementary Data 4), while blockers (after) with electrical stimulation alone (after) or electrical stimulation (after) with immediate vehicle addition (after) had no effect (Fig. 4d, e; e(ii): $n = 35$, $p = 0.9999$; e(iv): $n = 35$, $p = 0.8553$; Mann–Whitney test; Supplementary Data 4). Increases in neural activity of opsin-expressing populations is observed in response to CTZ, but not to vehicle application immediately following electrical stimulation, nor when CTZ or electrical stimulation are applied by themselves (Fig. 4e; yellow bars; immediate CTZ (after), $n = 10$, v/s immediate vehicle (after), $n = 35$; CTZ added later (after), $n = 18$; electrical stimulation (after), $n = 35$; $p < 0.0001$; Mann–Whitney test). These findings show that Interluminescence depends on presynaptic activity but that it is independent of the postsynaptic actions of synaptic neurotransmitters on neurotransmitter receptors.

**Interluminescence depends on presynaptic vesicle release.** To test whether presynaptic vesicle fusion is essential for Interluminescence, we used botulinum toxin (BoNT) to inhibit this process. BoNT, a neurotoxin that cleaves SNARE proteins, inhibits vesicular fusion and cargo release from presynaptic vesicles[18] (Fig. 5a). In BoNT-treated cultures, the level of luciferase in the media was reduced compared to untreated cultures, consistent with a decrease in vesicle release (Supplementary Fig. 5). CTZ added immediately following electrical stimulation of presynaptic neurons failed to trigger Interluminescence as indicated by the absence of postsynaptic activity in BoNT-treated cultures. This result is consistent with block of synaptic vesicle fusion and block of luciferase release into the synaptic cleft (Fig. 5b, c for individual traces and ladder plots; $n = 21$, electrical stimulation + CTZ $p = 0.7173$; Wilcoxon matched-pairs signed rank test; Supplementary Data 5). In contrast, direct application of blue light elicited robust spiking in opsin-expressing post-synaptic neurons ($n = 21$, blue light LED $p < 0.0001$; Wilcoxon matched-pairs signed rank test).

BoNT inhibits exocytosis of both small synaptic vesicles that contain small-molecule transmitters and large dense-core vesicles (LDCVs) that contain peptides. We used the sorting signal for the neuropeptide, POMC, to target luciferase to LDCVs using an hPOMC1-26-sbGLuc-eGFP fusion protein. We assessed the localization of sbGLuc-eGFP utilizing an antibody to dopamine $\beta$-hydroxylase which labels LDCVs[19] (Supplementary Fig. 6). We observed partial colocalization of eGFP and anti-dopamine $\beta$-hydroxylase, confirming the presence of sbGLuc in dopamine $\beta$-hydroxylase expressing synaptic vesicles, but we also observed sbGLuc-eGFP signals consistent with a considerable fraction of luciferases in other vesicles.

**Interluminescence-mediated activation of postsynaptic neurons requires opsins.** We performed two experiments to test if bioluminescence-mediated activation of postsynaptic neurons occurs through opsins. First, we inactivated the step-function opsin ChR2(C128S) by exposure to longer wavelength light (590 nm). This protocol resulted in complete opsin inactivation (Fig. 6a schematic, Fig. 6b upper trace). To test if this inactivation also blocked Interluminescence, we applied synaptic blockers and CTZ to spontaneously spiking neurons, resulting in a neuro-transmitter independent increase in spiking through biolumi-nescent activation of the opsin (Fig. 6b, lower trace). 590 nm light completely abolished spiking initiated by CTZ (Fig. 6b, lower trace) consistent with a need for recruitable opsins for Inter-luminescence (Fig. 6c ladder plot; SB + CTZ, $n = 51$, $p = 0.0369$; Green Light, $n = 51$, $p = 0.0001$; Wilcoxon matched-pairs signed rank test; Supplementary Data 6). Second, we used a

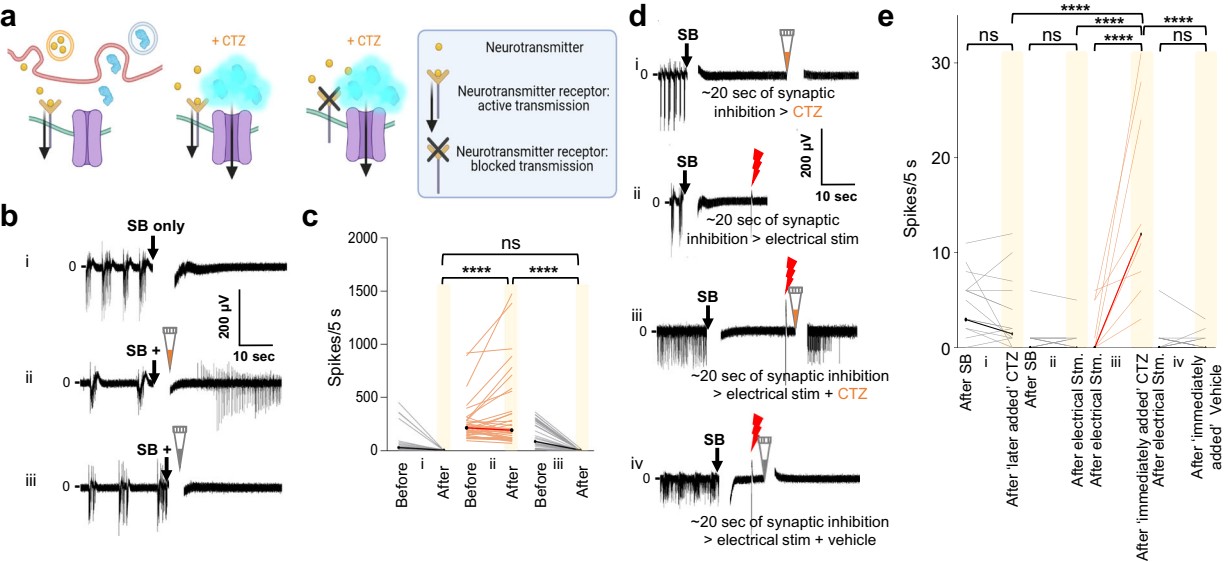

**Fig. 4 Interluminescence elicits postsynaptic firing increase in the presence of synaptic blockers dependent on presynaptic neuronal activity. a** Illustrations showing release of synaptic vesicle contents (neurotransmitters: yellow spheres, luciferases: blue enzymes) with spontaneous presynaptic activity inducing postsynaptic responses with transmitters alone (left panel), with transmitters and bioluminescent activation of opsins in the presence of CTZ (middle panel), and the effect of application of synaptic blockers (SB), allowing to isolate the effects of bioluminescence-mediated synaptic transmission (right panel). **b** Traces from representative electrodes of opsin (ChR2(C128S)) expressing population applying to the culture (i) synaptic blockers alone, (ii) synaptic blockers together with CTZ, or (iii) synaptic blockers together with vehicle. **c** Ladder plots of recordings under the conditions depicted in (**b**) from electrodes across opsin-expressing populations comparing number of spikes 5 s before and after (i) synaptic blockers alone, (ii) synaptic blockers together with CTZ, or (iii) synaptic blockers together with vehicle. (i) SB alone, $n = 27$, (ii) SB + CTZ, $n = 37$, (iii) SB + vehicle, $n = 38$; SB alone (after) v/s SB + CTZ (after), $p < 0.0001$; SB + CTZ (after) v/s SB + vehicle (after), $p < 0.0001$; SB alone (after) v/s SB + vehicle (after), $p = 0.7305$; Mann–Whitney test. **d** Traces from representative electrode recordings of opsin-expressing population applying to the culture synaptic blockers followed after ~20 s by application of (i) CTZ, (ii) electrical stimulation, and electrical stimulation together with either (iii) CTZ or (iv) vehicle. **e** Ladder plots of recordings under the conditions depicted in (**d**) across populations. (i), $n = 18$, SB (after) v/s CTZ added ~20 s later (after), $p = 0.4022$; (ii), $n = 35$, SB (after) v/s electrical stimulation (after) $p > 0.9999$; (iii), $n = 10$; electrical stimulation (after) v/s immediate CTZ treatment (after), $p < 0.0001$; (iv), $n = 35$, electrical stimulation (after) v/s immediate vehicle treatment (after), $p = 0.8553$; Mann–Whitney test. Significant increase in activity of opsin-expressing populations is observed only when CTZ is applied immediately following electrical stimulation and not when CTZ or electrical stimulation are applied by themselves nor when vehicle is applied immediately following electrical stimulation (yellow bars: immediate CTZ (after) ($n = 10$) v/s CTZ added ~20 s later (after), ($n = 18$); electrical stimulation (after), ($n = 35$); immediate vehicle (after), ($n = 35$); $p < 0.0001$; Mann–Whitney test. The artifacts due to addition of reagents in MEAs are overlaid by a vertical white bar in the recording traces (the white gap right after addition of SB, CTZ or vehicle). Artifacts due to electrical stimulation are visible under the red bolts. ns, not significant; ****$p < 0.0001$.

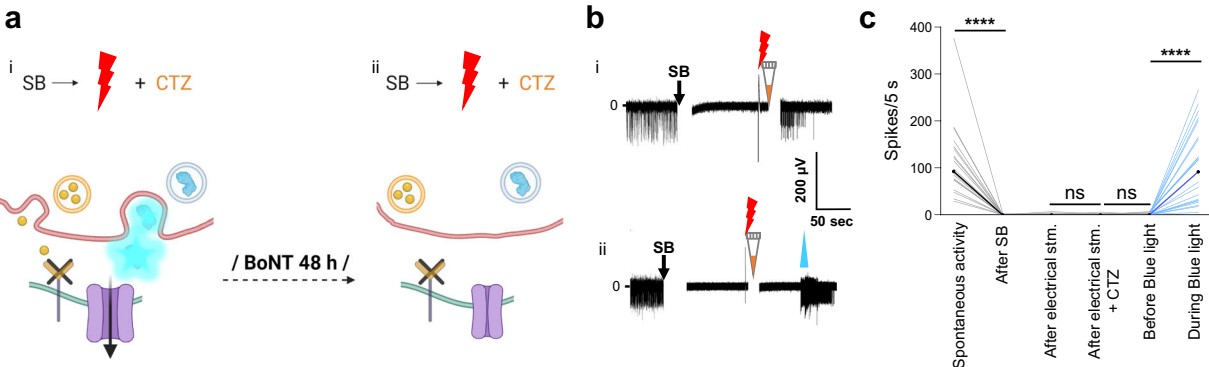

**Fig. 5 Interluminescence is dependent on presynaptic vesicle release. a** Schematics of synapses receiving synaptic blockers (SB) followed by electrical stimulation of presynaptic neurons together with CTZ application without (i) and after 48 h Botulinum Neurotoxin (BoNT) treatment (ii). **b** Representative trace of MEA recordings of opsin (ChR2(C128S)) expressing neurons without BoNT treatment (i, from Fig. 4diii) and 48 h after BoNT treatment of the co-culture (ii): electrical stimulation of presynaptic neurons together with CTZ application fails to elicit firing after BoNT treatment, while blue light still induces firing in the same recording. **c** Ladder plots of recording conditions as in (**b**) from electrodes across populations (comparisons are: spontaneous activity vs SB addition (after), $n = 21$, $p < 0.0001$; electrical stimulation (after) vs immediately following CTZ (after), $n = 21$, $p = 0.7173$; electrical stimulation + CTZ (after) vs blue light (before), $n = 21$, $p = 0.6055$; blue light (before) vs blue light (during), $n = 21$, $p < 0.0001$; Wilcoxon matched-pairs signed rank test). The artifacts due to addition of reagents in MEAs are overlaid by a vertical white bar in the recording traces (the white gap right after addition of SB or CTZ). Artifacts due to electrical stimulation are visible under the red bolts. ns, not significant; ****$p < 0.0001$.

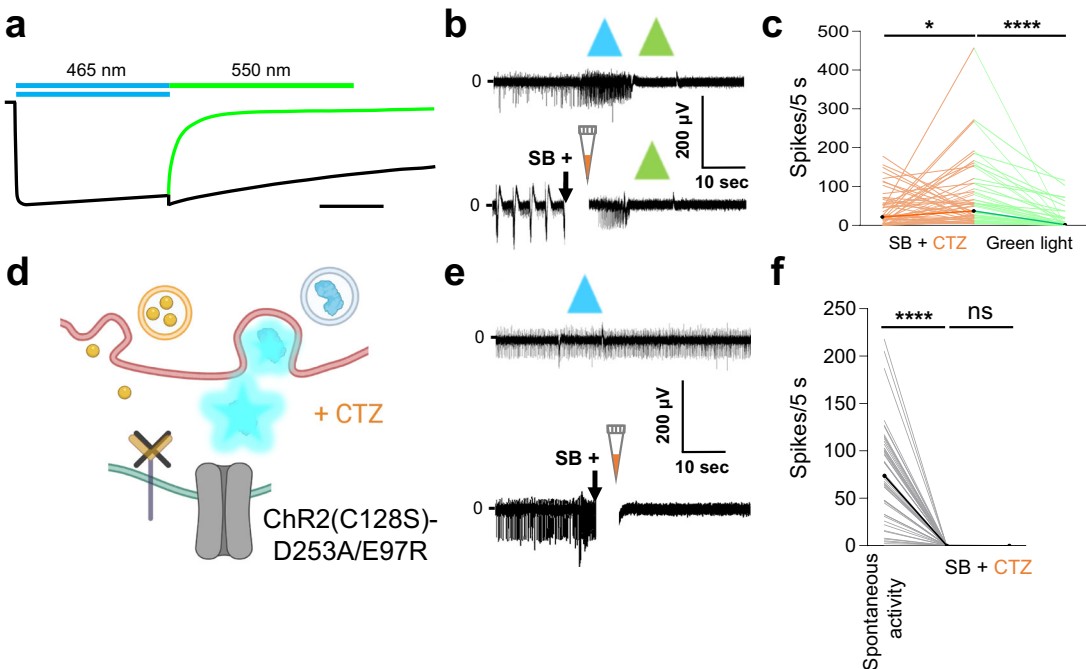

**Fig. 6 Interluminescence is mediated by bioluminescence activation of the opsin. a** Schematic showing the typical photocurrent of the step-function opsin ChR2(C128S) with a pulse of blue (465 nm) light (black trace under blue bar). There is prolonged depolarization after the light stimulation ends (continuation of black trace). Exposure to green (550 nm) light terminates the depolarization and returns the channel to its closed state (green trace under green bar; adapted from ref. [15], Fig. 2a; scale bar indicates 10 s). **b** Representative traces of MEA recordings of postsynaptic ChR2(C128S) expressing neurons from a cortical-striatal co-culture. ChR2(C128S) can be activated by blue light and inactivated by green light (upper trace). In the presence of synaptic blockers (SB) depolarization caused by CTZ is also inactivated by green light (lower trace), indicating that the channelrhodopsin mediates the Interluminescence effect. **c** Ladder plots depict recordings from electrodes across populations as in (b, lower trace) (SB + CTZ, $n = 51$, $p = 0.0369$; Green Light, $n = 51$, $p < 0.0001$; Wilcoxon matched-pairs signed rank test). **d** Schematic of synapse with postsynaptic neuron expressing a non-functional opsin, ChR2(C128S)-D253A/E97R. **e** Representative traces of MEA recordings of postsynaptic ChR2(C128S)-D253A/E97R expressing neurons from a cortical-striatal co-culture. Postsynaptic neurons expressing the mutant opsin show no responses to either direct blue light stimulation (upper trace) or to CTZ application (lower trace), indicating that Interluminescence is a specific effect through the opsin. **f** Ladder plots of recordings under the conditions depicted in (e, lower trace) (Before vs after addition of SB, $n = 49$, $p < 0.0001$; Wilcoxon matched-pairs signed rank test; SB alone (after), $n = 21$, v/s SB + CTZ (after: for non-functional opsin), $n = 49$, $p = 0.7870$; Mann–Whitney test). The artifacts due to addition of reagents in MEAs are overlaid by a vertical white bar in the recording traces (the white gap right after addition of SB + CTZ). ns, not significant; *$p < 0.05$; ****$p < 0.0001$.

non-functional opsin mutant ChR2(C128S)-E97R-D253A that does not produce photocurrent[20] (Fig. 6d schematic, Fig. 6e upper trace). In cultures expressing inactive ChR2(C128S)-E97R-D253A in postsynaptic neurons, CTZ generated bioluminescence, but no increase in spiking (Fig. 6e, lower trace, Fig. 6f ladder plot; SB alone (after), $n = 21$, v/s SB + CTZ (after), $n = 49$, $p = 0.7870$; Mann–Whitney test; Supplementary Data 6). These data indicate that Interluminescence is mediated by photocurrent generation following bioluminescent activation of the opsin.

**Interluminescence in vivo: induction of cell-partner specific brain dynamics.** An Interluminescence approach holds substantial distinct advantages for understanding behavior, as the complex processes underlying activities such as choice, memory and selective sensory processing inherently depend on cell-type-specific interactions between multiple brain areas. These interactions relay specific signals and create the dynamic states that facilitate or suppress specific channels of information. Such inter-areal processing is highly dependent on the specific type of neurons engaged in each area.

A prominent example of this kind of cell-type-specific dynamic is gamma oscillations, rhythmic patterns of activity (~30–100 Hz) that are predictive of successful sensory processing[21–23], and are believed to amplify the relay of sensory neural signals. Neocortical gamma depends on recruitment of fast-spiking, parvalbumin-positive (FS/PV) interneurons, either through endogenous or artificially applied glutamatergic drive, or by selective optogenetic activation[24–26]. Attentional gating of gamma in a given neocortical area is believed to be caused by excitatory intracortical[27,28] or thalamocortical[29–31] projections that recruit local FS/PV dynamics.

Given the potential utility of Interluminescence for in vivo studies, a crucial test is whether it can change network dynamics created by long-range, cell-type-specific communication. To test in vivo efficacy, we expressed the transmitting hPOMC1-26-sbGLuc in glutamatergic thalamic neurons (including ventral posterior medial and posterior medial) that target primary somatosensory neocortex (SI), including direct synaptic input to FS/PV[32,33]. The receiver excitatory opsins (ChR2(C128S/D156A)) were expressed under PV-Cre mediated control in SI FS/PV[24] (Fig. 7a and Supplementary Fig. 7; this subset of mice will be referred to as Opsin (+)). In SI superficial and granular layers, PV is nearly exclusively FS-type[24]. In a control group, we expressed hPOMC1-26-sbGLuc in thalamus but not the excitatory opsin in neocortex (Opsin (−)).

In both Opsin (+) mice and Opsin (−) control mice CTZ presentation caused a rise in bioluminescent light production (Fig. 7b; Supplementary Data 7). In Opsin (+) mice, we observed robust, broad-band gamma oscillation emergence that initiated with CTZ presentation (Fig. 7c; Supplementary Data 7). In contrast, similar changes were not observed in the Opsin (−)

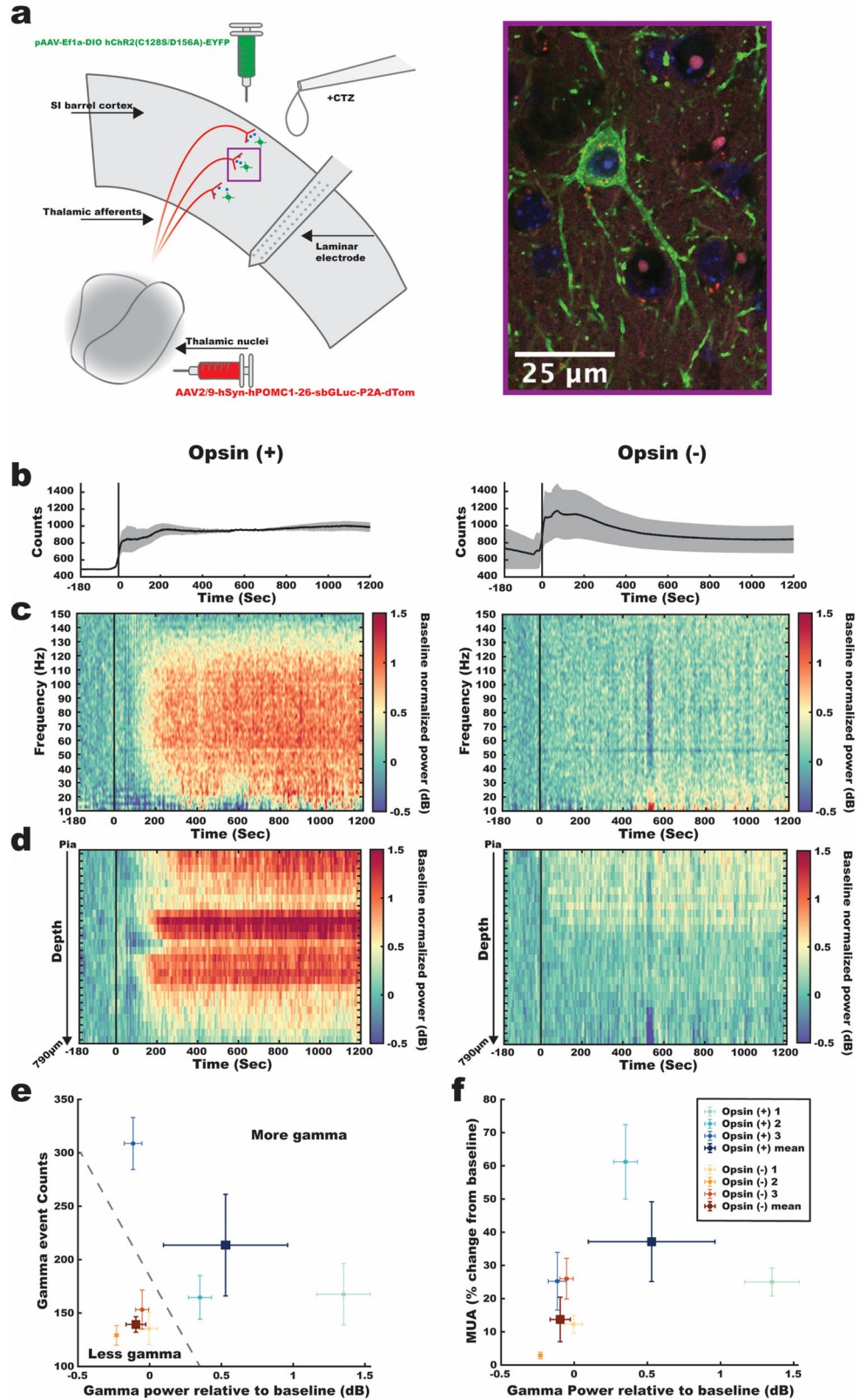

control group, despite robust bioluminescent photon output. These Interluminescence gamma increases were localized to more superficial and granular layers across the neocortical depth and were evident on ~half of all electrode contacts in the Opsin (+) group (Fig. 7d: Opsin (+) = 15.7 ± 10.1 SD electrodes/mouse; Opsin (−): M = 2.7 ± 2.5 SD; Supplementary Data 7).

To systematically quantify these differences, in each mouse in the Interluminescence and control groups we calculated the overall increase in gamma power on each electrode compared to baseline for the period 0–1200 s after bioluminescent signal onset, and the probability of significant gamma events defined as ≥ 100 ms of increased gamma (see "Methods" for details). As shown in Fig. 7e,

**Fig. 7 Modulation of postsynaptic neural activity by Interluminescence in vivo. a** Schematic of the in vivo Interluminescence configuration targeting somatosensory thalamic nuclei and SI Barrel cortex (left panel). Confocal image of a PV cell expressing the excitatory step-function opsin with an EYFP tag (green) along with thalamocortical axon terminals expressing the hPOMC1-26-sbGLuc with dTomato tag (red) and cell nuclei stained with DAPI (blue) (right panel). **b** The average bioluminescence signal with ±1 SEM in semi-opaque bands for the Opsin (+) and Opsin (−) group ($n = 2$ and $n = 3$ animals, respectively). The bioluminescence time series were obtained by averaging across a circular ROI (50 pixel/290 μm diameter) adjacent to the electrode insertion point in the acquired image series. **c** Time-frequency spectrograms averaged across all laminar electrode contacts for the Opsin (+) (left panel) and Opsin (−) (right panel) cohorts. Time zero refers to the onset of the bioluminescence signal. **d** Depth profiles of gamma-band power (dB) (80–100 Hz) across the electrode contacts for the Opsin (+) group (left panel) and Opsin (−) group (right panel). **e** Gamma-band power (dB) relative to baseline plotted against gamma event counts after baseline. Small circles are individual animals, averaged across the electrode contacts. Error bars are ±1 SEM across the electrode contacts for a given animal. Large squares are the mean values for the Opsin (+) group and Opsin (−) group. Error bars represent ±1 SEM across animals. $N = 3$ animals for each group. **f** Gamma-band power (dB) relative to baseline plotted against MUA percent change from baseline. Same conventions as in (**e**). $N = 3$ animals for each group.

the three Opsin (−) mice showed no consistent increase in either overall strength or likelihood of gamma expression. In contrast, the Opsin (+) mice showed robust increases in one or both measures. Analysis of each metric at the mouse or electrode level showed significantly higher values for the Opsin (+) group (Fig. 7e Mean gamma (dB) power across electrodes: Opsin (+) Mean = 0.53 ± 0.86 SD, Median = 0.34; Opsin (−) Mean = −0.10 ± 0.24 SD, Median = −0.17; KS test $p < 0.0001$; 1-tailed Mann–Whitney $U$ test $U = 4163$, $n_1 = n_2 = 75$, $p < 0.0001$; Mean gamma event number across electrodes: Opsin (+) Mean = 213.60 ± 139.53 SD, Median = 222; Opsin (−) Mean = 139.28 ± 72.97 SD, Median = 149; KS test $p < 0.0001$; 1-tailed Mann–Whitney $U$ test $U = 3609.50$, $n_1 = n_2 = 75$, $p < 0.0001$; Mean gamma power across mice Opsin (+) Mean = 1.16 ± 1.09 SD, Median = 0.88; Opsin (−) Mean = 0.03 ± 0.18 SD, Median = 0.09; 1-tailed Mann–Whitney $U$ test $U = 9$, $n_1 = n_2 = 3$, $p < 0.05$; Mean gamma event number across mice Opsin (+) Mean = 327.25 ± 45.58 SD, Median = 304.25; Opsin (−) Mean = 199.33 ± 36.08 SD, Median = 203.75; 1-tailed Mann–Whitney $U$ test $U = 9$, $p < 0.05$; Supplementary Data 7; see "Methods" for details).

In many cases, such as during the allocation of attention, increased local action potential firing is closely tied to increases in gamma-band activity[21]. Organization of FS/PV activity into a gamma pattern is believed to enhance signal relay by creating windows of opportunity for increased local firing, and increased firing due to rebound excitation[34]. Further, increased spiking associated with gamma increases may also reflect the disproportionate contribution of FS to multi-unit activity (MUA) measures, as FS/PV firing rates are typically an order of magnitude higher than those of nearby pyramidal neurons[35,36]. That said, FS/PV evoke powerful, soma-targeted inhibition, and robust recruitment of this cell class can create local suppression of spiking[32,35], e.g., through optogenetic FS/PV recruitment at higher light intensities[24].

As shown in Fig. 7f, during CTZ presentation, Opsin (+) mice showed MUA increases from baseline that were higher than, or equivalent to, the highest mean values of Opsin (−) mice. However, only one mouse showed a significant separation from the group. Accordingly, MUA differences were significant when analyzed at the electrode, but not the mouse, level (Fig. 7f *Mean MUA (percent change from baseline) across electrodes*: Opsin (+) Mean = 37.14 ± 45.50 SD, Median = 15.49; Opsin (−) Mean = 13.71 ± 21.44 SD, Median = 8.96; KS test $p < 0.0001$; 2-tailed Mann–Whitney $U$ test $U = 3899$, $n_1 = n_2 = 75$, $p < 0.0001$; Mean MUA (percent change from baseline) across mice: Opsin (+) Mean = 77.72 ± 47.40 SD, Median = 52.15; Opsin (−) Mean = 27.15 ± 18.38 SD, Median = 27.24; 2-tailed Mann–Whitney $U$ test $U = 9$, $p > 0.05$; Supplementary Data 7).

## Discussion
The present study shows that Interluminescence can provide a robust, synapse-selective, activity-dependent control of neural connectivity between specific pre- and postsynaptic partners. We show that presynaptic luciferase can activate postsynaptic opsins following presynaptic activity and only if luciferin is present. This chemogenetic element of the strategy provides an additional level of experimenter control. Interluminescence is modular, as light emission from the presynaptic luciferase can, in principle, activate any postsynaptic photoreceptor, including excitatory and inhibitory opsins and light-sensing GPCRs. Further, the approach is highly specific in that Interluminescence does not seem to have a volumetric effect.

Given its features, Interluminescence has the potential to provide a platform technology, in which the activity-dependence and postsynaptic impact of Optical Synapse recruitment can be selected. In this first instantiation described here, to maximize the coupling efficiency between luciferase and opsin, we used *Gaussia* luciferase, sbGLuc, which has high light emission, and the step-function opsin ChR2(C128S) and the anion channelrhodopsin 2 from *Guillardia theta*, hGtACR2 which have high photon sensitivity[6,8,9,20]. The precise photon density required to create the Interluminescence effects we observed is difficult to quantify in the abstract, as it depends on numerous factors including luciferase density, the impact of biologically specific variables in the cleft (e.g., pH sensitivity) and specific details of synaptic connectivity (e.g., synaptic distance, number of Interluminescent synapses expressing both components, location of these synapses on the postsynaptic cell, etc.). The luciferase–opsin combinations used here provide a baseline proof of concept that ample photon production was achieved in these in vitro and in vivo conditions, from a point of comparison for future luciferase–opsin pairings.

A prerequisite for Interluminescence is close proximity of the light emitter release presynaptically and light sensor located in the postsynaptic cell. In the instantiation of Interluminescence described here, we chose to express luciferase in synaptic vesicles, to concentrate light-producing enzymes to the presynaptic active zone and to ensure that presynaptic activity was required for luciferase release. We were able to take advantage of well-characterized pharmacological tools to manipulate vesicle exocytosis, including Botulinum Toxin (BoNT), thereby testing several assumptions including that Interluminescence should dependent on presynaptic vesicle fusion. BoNT inhibits both small synaptic vesicles and LDCVs[37] and immunohistochemistry revealed colocalization of luciferases in both dopamine β-hydroxylase-containing and non- dopamine β-hydroxylase-containing vesicles[19]. Thus, although we used the POMC sorting signal to concentrate luciferase in peptide-containing LDCVs, our results suggest that luciferase was present in LDCVs as well as non-peptide-containing synaptic vesicles. Achieving a higher degree of specific targeting of luciferases to specific vesicles, either LDCVs or small synaptic vesicles could provide a way to establish functional connectivity via Interluminescence based on the stimulation frequency. For example, there is evidence that SVs are

preferentially released in response to low-frequency stimuli compared to LDCVs which are preferentially released in response to higher frequency stimuli[38].

Synaptic vesicles are located in different subcellular domains of neurons including at presynaptic active zones, soma, dendrites, and axons. It is, therefore, possible and likely that luciferase is released at multiple sites following neuronal depolarization. Interluminescence likely reflects optical signaling at functional presynaptic synapses[39] because of the need for close proximity of luciferase and postsynaptic opsins across a shared synaptic cleft. Interluminescence outside of bona fide synapses is unlikely although this possibility could be explored in the future. Further, given that 20 s after release addition of luciferin does not elicit a postsynaptic response, it is most likely that luciferases diffused away from the synaptic space do not have a photon density high enough to activate opsins along the neuron. Thus, in contrast to neuropeptide transmission Interluminescence does not seem to have a volumetric effect.

The Interluminescence method described here is complementary to recently described orthogonal neuropeptide–receptor systems which regulate communication between genetically targeted pre- and postsynaptic partners. One is based on the insect peptide allatostatin and its receptor, both of which are inert in mammals[40]. The allatostatin receptor links via Gi/o-proteins to inhibit cyclic adenosine monophosphate (cAMP) and activate G-protein-coupled inward-rectifier potassium (GIRK) channels. Activity-dependent release of biologically active allatostatin from presynaptic neurons induces inhibition of allatostatin receptor-expressing subpopulations of postsynaptic neurons. The other system uses a *Hydra* derived presynaptically expressed neuropeptide and a matching postsynaptic cation channel that is opened by the peptide[41]. Upon activity-dependent presynaptic peptide release this heterologous synapse creates novel calcium fluxes postsynaptically and resulting in neural activation. Interluminescence, however, has some distinct advantages. First, Interluminescence is highly modular; luciferase-emitted light can be used to activate or inhibit partnering neurons depending on the opsin expressed, an advantage over an approach that requires separate systems for activation and inhibition. Second, Interluminescence utilizes opsins as universal current conductors, effecting direct changes in the membrane potential of the post-synaptic partner, an advantage over GPCR signaling pathways or $Ca^{2+}$ flux, both of which have the potential to engage a multitude of intracellular events. Third, transmission via synthetic chemical synapses is not highly restricted to presynaptic location, consistent with neuropeptide volume transmission. In contrast, luciferase-dependent light emission decays over time and luciferases that diffuse beyond the synaptic cleft are unlikely to activate post-synaptic neurons. Fourth, synthetic chemical synapses are always on, whenever the presynaptic cell is active, and they are not under temporal control. By contrast, Interluminescence can be temporally gated by controlling luciferin availability, a feature advantageous for assessing the behavioral impact.

In summary, Interluminescence provides a unique technology for interrogating specific neural circuits with substantial temporal and spatial control. Interluminescence can boost or down regulate synaptic efficacy at specific synapses, it can be used to bias synaptic output, e.g. from inhibitory to excitatory and vice versa, and, in principle, it can establish new functional synaptic connections for example from silent to active. With rapid advances in the available palette of luciferases and opsins, this strategy can expand to meet a wide array of experimental needs.

## Methods

**Materials**. The luciferase substrate, coelenterazine (CTZ), was purchased from NanoLight Technology (Pinetop, AZ): Coelenterazine free base, the natural form of CTZ (NanoLight # 303), was dissolved at 50 mM in NanoFuel (NanoLight # 399);

CTZ was further diluted 1:50 in culture medium for a 1 mM working solution that was further diluted 1:100 when added to MEAs for a final concentration of 10 μM. The same dilutions were carried out with just NanoFuel for vehicle. Cocktail of synaptic blockers included NBQX (abcam # ab120046), D-AP5 (abcam # ab120003), Gabazine (Sigma Aldrich # S106), CGP 55845 (Sigma Aldrich # SML0594) and Strychnine (Sigma Aldrich # S0532). Botulinum Neurotoxin (BoNT/A1) was purchased from Metabiologics (Madison, Wisconsin).

**Plasmids**. The coding sequence for the N-terminal tagged luciferase construct with the leader peptide (amino acids 1-26) from the human *pro-opiomelanocortin* gene (hPOMC1-26)[11,12], the *Gaussia* luciferase variant sbGLuc[10], a P2A self-cleaving peptide, and the dTomato sequence[42] was synthesized (Genscript) and cloned into pcDNA3.1-CAG and pAAV-hSyn to generate pcDNA3.1-CAG-hPOMC1-26-sbGLuc-P2A-dTomato and pAAV-hSyn-hPOMC1-26-sbGLuc-P2A-dTomato. Removal of P2A-dTomato and replacement by the coding sequence for EGFP generated pcDNA3.1-CAG-hPOMC1-26-sbGLuc-EGFP. Generation of pcDNA3.1-CAG-ChR2(C128S)-EYFP and its non-functional mutant pcDNA-CAG-ChR2(C128S)-E97R-D253A-EYFP are described in detail in Berglund et al. 2020[20]. The coding sequence for hGtACR2-EYFP was cloned into pcDNA3.1-CAG from pFUGW-hGtACR2-EYFP (a gift from John Spudich; Addgene plasmid # 67877; RRID:Addgene_67877).

**Virus**. High titer stocks of AAV2/9-hSyn-hPOMC1-26-sbGLuc-P2A-dTomato were made in-house using triple plasmid transfection in HEK293 cells (Agilent, Cat # 240073-41)[8]. Briefly, subconfluent HEK293 cells grown in 10 cm culture dishes were transfected with 24 μg of the helper plasmid pAd delta F6, 20 μg of the serotype plasmid AAV2/9, and 12 μg of the pAAV-hSyn-hPOMC1-26-sbGLuc-P2A-dTomato plasmid using Lipofectamine 2000. Virus was purified from cells and supernatant after 72 h through an aqueous two-phase system. Virus was dialyzed against PBS (w/o Ca, Mg) overnight at 4 °C, followed by concentration in Amicon Ultra-0.5 mL Centrifugal Filters. Viral titers were determined by Q-PCR for the WPRE element. Larger quantities of virus were made by ViroVek.

### In vitro

*Primary neurons*. Primary neurons harvested from embryonic day 18 (E18) rat embryo cortex, hippocampus or striatum of both sexes were obtained from BrainBits, LLC, and processed according to the protocol provided by the company. Briefly, tissue was incubated for 10 min at 30 °C in Hibernate E (minus calcium and B27 supplement; HEB, BrainBits) containing 2 mg/ml papain (BrainBits). Papain solution was removed, replaced by HEB medium, and tissue was triturated for about 1 min (90% tissue dispersal) using a 9″ sterile silanized glass Pasteur pipette (BrainBits), avoiding air bubbles. Undispersed pieces were allowed to settle for 1 min before the supernatant was transferred to a sterile 15 ml tube and spun at 1500 rpm for 10 min to collect the cell pellet. The pellet was resuspended in pre-warmed and equilibrated NbActiv1 medium (BrainBits) and the cells were counted by Hemocytometer using Trypan blue stain.

*Nucleofection*. Nucleofection of E18 primary rat neurons was carried out per manufacturer's instructions (Amaxa Rat Neuron Nucleofector Kit # VPG-1003). Briefly, $1 \times 10^6$ primary neurons were collected and resuspended in 100 μl of Nucleofector Solution at room temperature. The cell suspension was combined with 1 μg plasmid DNA and transferred to the nucleofection cuvette. The Nucleofector 2b Device (LONZA # AAB-1001) was used for nucleofection with Nucleofector Program G-013.

*Neuron culture on MEAs*. For the mixed culture set-up on MEAs, cortical neurons nucleofected with either the luciferase construct (hPOMC1-26-sbGLuc-P2A-dTomato) or the opsin construct (ChR2(C128S)-EYFP or hGtACR2-EYFP) were mixed at a 1:1 ratio and were plated on the electrode area ($1 \times 10^5$ cells/10 μl/well) of 1-well MEA dishes (60MEA200/30iR-Ti; Multi Channel Systems, Germany) coated with 0.1% polyethyleneimine (Sigma # P3143) and 50 μg/ml laminin (Gibco # 23017-015) in culture medium consisting of Neurobasal Medium (Gibco # 21103-049), B-27 supplement (Gibco # 17504-044), 2 mM Glutamax (Gibco # 35050-061), and 5% Fetal Calf Serum (FCS). The following day, the medium was replaced with serum-free medium (NB-Plain medium). Half of the medium was replaced with fresh NB-Plain medium every 3–4 days thereafter. For the co-culture set-up, neurons nucleofected with either the luciferase construct (hPOMC1-26-sbGLuc-P2A-dTomato) or the opsin construct (functional opsin ChR2(C128S)-EYFP or non-functional opsin ChR2(C128S)-E97R-D253A-EYFP) were plated in separate compartments of a 2 well silicon insert (Ibidi # 80209, Germany), placed on the MEA electrodes in such a way that the total number of electrodes were approximately divided equally between the two populations. Once neurons were attached, after ~18 h, the insert was removed and the populations were allowed to establish synaptic connections. Half of the medium was replaced with fresh NB-Plain medium every 3–4 days.

*MEA recordings*. MEA2100-Lite-System (Multichannel Systems, Germany) was used for all MEA recordings. Consistently spiking neurons were used for record-ings between DIVs 14–25 for the mixed and co-culture set-ups; only cultures

showing spontaneous electrophysiological activity were used. All-*trans* retinal (R2500; Sigma-Aldrich, St. Louis, MO) was added to the culture medium to 1 µM final concentration before electrophysiological recordings. Prior to recording, all reagents were pre-warmed to 37 °C. MEAs were transferred from the $CO_2$ incubator to the heated MEA2100 head stage maintained at 37 °C, and the cultures were allowed to equilibrate for 5–10 min. The head stage was situated on a microscope stage (Zeiss Observer 1) with a fluorescent light source, allowing light stimulation of cultures at different wavelength through the objective. A micropipette was used to add reagents with the reagent drop gently touching the liquid surface, creating a time-locked artifact in the recordings. Recordings were carried out with a sample rate of 10,000 Hz. After recording, the media in the wells was replaced with fresh pre-equilibrated and pre-warmed NB-Plain media, and cultures were used for another round of recording the next day. MC Rack software was used for data acquisition. All MEA analysis was done offline with MC Rack software (Multi-channel Systems; RRID: SCR_014955) and NeuroExplorer (RRID: SCR_001818). Spikes were counted when the extracellular recorded signal exceeded 9 standard deviations of the baseline noise. For assessing the effects of CTZ (10µm final concentration), only electrodes displaying the expected change in spiking activity with blue light from the fluorescent light source, i.e. opsin-expressing neurons, were evaluated. Pooled data were obtained from different electrodes (a) of the same culture, (b) from different cultures, and (c) over different DIVs.

*Electrical stimulation.* Electrical stimulation on MEA co-cultures was carried out using the integrated stimulus generator in the MEA head stage (MEA2100 Stimulator). The burst stimulation pattern was selected for a 100 µA current stimulus train, with the inter-pulse interval of 10 ms, and the pulses within this train were repeated 5 times.

*Un-cut vs Cut experiment.* Co-cultures were allowed to mature until there was synchronous firing activity across the co-culture. Effects of blue light, current stimulation, and CTZ were recorded from these synaptically connected un-cut co-cultures. Thereafter, in the same co-cultures, inter-population connections were severed by running a piece of thin silicon like an eraser along the midline between the two populations. These cut co-cultures, which had lost the inter-population synchronicity, were then subjected to the same treatments (blue light, current stimulation and CTZ).

*Synaptic blockers.* The cocktail of synaptic blockers (SB) (final concentrations indicated) included NBQX (10 µM), D-AP5 (50 µM), Gabazine (100 µM), CGP55845 (100 µM) and Strychnine (1 µM). Aliquots were stored at −80 °C and each time thawed freshly right before the start of the MEA recording. The SB cocktail was incubated at 37 °C before being added to the MEA and was added gently as 10 µl drop to the neuronal media in the MEA well. For recordings involving a mixture of SB cocktail with either CTZ or vehicle, CTZ or the vehicle stocks were freshly diluted with the SB cocktail to attain the final CTZ concentration of 10 µM or equivalent in case of the vehicle.

*Botulinum neurotoxin (BoNT).* BoNT/A1 was used as a blocker for the vesicular release of the presynaptic luciferase. BoNT was used at 30 ng/ml for 48 h before recording experiments.

*Confocal Imaging.* For confocal microscopy nucleofected cortical neurons were grown on Poly-D-Lysine coated coverslips (Neuvitro GG-12-PDL) in 24-well dishes until DIV 21. Neurons were fixed by completely removing the media from each well and then adding 500 µL of 4% paraformaldehyde and incubating for 15 mins at RT, followed by 3 washes for 5 min each in PBS. Neurons were permeabilized by incubating in 0.1% Triton X-100 and again were washed 3 times for 5 min each in PBS. Neurons were blocked with 1% Bovine Serum Albumin (BSA) in PBST (PBS + 0.1% Tween 20) for 1 hr, incubated for 12 h at 4 °C with a rabbit polyclonal anti-Dopamine β Hydroxylase (DβH) antibody (Millipore Sigma, AB 1585, diluted 1:2000 in 1% BSA in PBST), then washed 3 times, for 5 min each time, with PBST. Neurons were then incubated for 1 hr at RT with Donkey anti-Rabbit IgG H&L (Alexa Fluor 594; ab150076; diluted 1:500 in 1% BSA in PBST) and washed 3 times for 5 min each in PBST. Cells were mounted in antifade mounting media (Vectashield Hardset, H-1500-10) containing DAPI and imaged with a Nikon A1 confocal laser scanning inverted microscope using a Nikon Plan Apo VC 60x/1.40 Oil DIC N2 objective (1024 × 1024 µm). To image sbGLuc-eGFP the optical sections were scanned with the 561 nm laser line at 60% intensity. To detect DβH+ dense core vesicles with Alexa Fluor 594 the 561 nm laser line at 2% intensity was utilized. Detection was done with a 450/50 filter cube for eGFP and 595/50 for Alexa Fluor 594. The raw images were exported as TIF files and analyzed with ImageJ (Rasband, W.S., ImageJ, U.S. National Institutes of Health, Bethesda, Maryland, USA, http://imagej.nih.gov/ij/).

*Statistics and reproducibility.* All analyses were performed with Prism software (GraphPad 8.2.1; San Diego, CA), which provides the evaluation of the suitability of the test for the specific data set. MEA data were collected from multiple recordings within each experiment and from multiple experiments. For randomization, the plates were switched for different treatment conditions, e.g., the plate

used for CTZ treatment on one day was used for vehicle treatment the next day and vice versa. Positive control blue light treatment using the fluorescent light source was done for each recording along with other treatments as the basis for selecting electrodes for analysis of opsin-expressing postsynaptic neurons. All the treatments (e.g. CTZ vs vehicle) were carried out on similar DIVs (13–21 for mixed cultures; 26–29 for co-cultures) to control for age and synaptic connectivity-related variations within neuronal cultures on MEAs. For each MEA data ladder plot, $n =$ number of electrodes were assessed. The differences in number of spikes before and after treatment were assessed for significance. Due to non-normal distribution of data, non-parametric paired Student's $t$ tests (two-tails) were used. To evaluate the within-group differences, Wilcoxon matched-pairs signed-rank test was used, and to evaluate across-groups differences, Mann–Whitney U tests were used with significance set at $p < 0.05$ (ns, not significant; *$p < 0.05$; **$p < 0.002$; ***$p < 0.0002$; ****$p < 0.0001$) using 95% confidence level. Throughout the paper, the medians are highlighted for each ladder plot in the figures. The time analyzed for the number of spikes before and after treatment was 5 s for all ladder plots, removing the time pertaining to artifacts due to the addition of reagents (indicated by the white gap area in the representative recording traces as noted in the figure legends). $n$ and $p$ values and the type of statistical test used are described for each ladder plot in the figure legends and results.

## In vivo

*Animals.* Six PV-Cre mice (all male; JAX stock #008069) aged 9 to 19 weeks ($M = 16.10$, $SD = 3.72$) were used. Three mice were injected with the luciferase virus (AAV2/9-hSyn-hPOMC1-26-sbGLuc-P2A-dTomato) in somatosensory thalamus along with injections of a Cre-dependent excitatory step-function opsin in SI (Opsin (+) animals). As a control, three additional mice were also injected with the luciferase virus in somatosensory thalamus, but no opsin was introduced (Opsin (−) animals). The Opsin (−) animals thus should produce light in SI due to thalamocortical projections to SI, but no optogenetic effect ought to occur because of the absence of a postsynaptic opsin. Imaging data are unavailable for one of the Opsin (+) animals due to a software malfunction. Mice were housed in a vivarium on a reversed light-dark cycle and had free access to food and water. All procedures were conducted in accordance with all relevant ethical regulations for animal testing and research and under a study protocol approved by the Animal Care and Use Committee of Brown University.

*Surgeries and course of experiment.* Approximately three weeks prior to the day of the experiment, each animal was anesthetized (~1% isoflurane), fitted with a steel headpost, and injected with viral constructs via burr holes made with a dental drill. Animals received an injection of 400 nl of luciferase virus (AAV2/9-hSyn-hPOMC1-26-sbGLuc-P2A-dTomato) into the somatosensory thalamus (−1.75 A/P ± 0.05, M/L 1.575 ± 0.175, D/V −3.4 relative to Bregma). This injection strategy targeted somatosensory thalamus broadly, likely infecting neurons in both the ventral posterior medial and the posterior medial nuclei. All viral injections were performed through a glass pipette fitted in a motorized injector (Stoelting Quintessential Stereotaxic Injector, QSI). The Opsin (+) animals also received additional viral injections of 200 nl of the excitatory step-function opsin (pAAV-Ef1a-DIO hChR2(C128S/D156A)-EYFP; a gift from Karl Deisseroth; Addgene viral prep # 35503-AAV1; RRID:Addgene_35503) in three locations of SI equidistantly spaced around a central SI point (−1.25 A/P, 3.25 M/L) at a depth of 350 µm. All viral constructs were delivered at a rate 50 nl/min.

After 2–3 weeks of recovery, and to allow for viral expression of the constructs, experiments were conducted under isoflurane at ~1% (0.5–2%). A dental drill was used to make a 3 mm diameter circular craniotomy centered over SI (−1.25 A/P and 3.25 M/L relative to Bregma). The exposed brain remained covered in saline throughout the experiment. The animal was moved to a light tight and electrically shielded box and continued to receive anesthesia. A 32-channel probe was inserted into the cortex perpendicularly to the cortical surface at a rate of ~10 µm/s using a motorized micromanipulator (Siskiyou MD7700) to a depth of 795 µm or until the highest contact on the probe disappeared from view into the cortical tissue as viewed from a stereoscope. The probe was then allowed to rest in this position for ~30 min before starting the experiment. After baseline recordings of a minimum of 3 minutes, the luciferin CTZ was introduced and recordings continued for a minimum of 20 minutes. At the conclusion of the experiment, mice were euthanized with isoflurane and perfused transcardially with 4% paraformaldehyde (PFA). The brain was removed and post fixed in 4% PFA at 4 °C for approximately 48 h after perfusion. The brain was then placed in 30% sucrose at 4 °C for a minimum of 36 h before slicing. Brains were then sectioned at 50 µm on a cryostat (Leica CM30505) and mounted on glass slides. Fluorescent tags in the sectioned brains were imaged on a Zeiss LSM 800 confocal microscope to verify correct viral targeting.

*Luciferin delivery.* Water soluble coelenterazine (Nanolight #3031) was dissolved in sterile water (1 µg/ml) to yield a final concentration of 2.36 mM. The solution was loaded into a 250 µl glass syringe (Hamilton #80701) fitted with a ~1 cm length of 18-gauge plastic tubing. The Hamilton syringe and tubing were placed in a motorized injector (Stoelting Quintessential Stereotaxic Injector, QSI). The tip of the plastic tubing was lowered into the pool of saline over the craniotomy using a

micromanipulator until it touched the surface of the skull. The tip of the tubing was further adjusted so that it rested at a distance of ~3 mm from the opening edge of the craniotomy. The luciferin CTZ was delivered by infusing 50 µl of the solution into the saline over the open craniotomy at a rate of 25 µl/min.

*Imaging and electrophysiological recordings.* Electrophysiological data were acquired using an Open Ephys acquisition board (http://www.open-ephys.org/) connected via an SPI interface cable (Intan) to a 32-channel head stage (Intan). A 32-channel laminar probe (Neuronexus, A1x32-Poly2-5mm-50s-177) was connected to the head stage. The iridium electrode contacts on the probe covered a linear length of 790 µm and were arranged into two columns of 16 contacts spaced 50 µm apart. The data were acquired using the Open Ephys GUI software at a sampling rate of 30 kHz and referenced to a supra-dural silver wire inserted over the right occipital cortex. Imaging data were acquired using an Andor iXon Ultra 888 EMCCD camera attached to a Navitar Zoom 6000 lens system. The data were acquired using Andor Solis data acquisition software (Andor Solis 64 bit, v4.31). The field of view was centered over the craniotomy and adjusted to encompass the full diameter of the craniotomy. Images (512 × 512 pixels, ~6 µm²/pixel) were acquired continuously at an exposure length of 1 s and an electron multiplication gain of 300. The data were acquired in units corresponding to the number of electrons recorded by a given pixel. A TTL pulse synchronized the recording of the imaging and electrophysiological data.

*Electrophysiology analysis.* Offline analyses of both electrophysiological and imaging data were performed in MATLAB R2020a (The Mathworks Inc.). The electrophysiological data were down sampled to 10 kHz. For each recording, electrode contacts with RMS values more than three times the interquartile range above the 3rd quartile or three times the interquartile range less than the 1st quartile of all 32 electrode contacts were marked as errant and removed from further analyses. Across all animals a total of seven electrode contacts were marked bad, so all reported electrophysiological data are from the remaining 25 contacts. These electrode contacts were then re-referenced to the common average reference[43].

For the time-frequency analysis of the local field potential, the data were further down sampled to 1 kHz and high-pass filtered with a cutoff at 1 Hz (3rd order Butterworth). Spectral analysis of the time series of each electrode contact was performed using a sliding multitapered fast Fourier transform using the Chronux software package for MATLAB (version 2.11, http://chronux.org)[44]. The time-bandwidth product for the multitaper analysis was set to 3 and 5 tapers were used. Sliding windows of 10 s in steps of 1 s were employed to analyze the spectro-temporal evolution of the time series, and each 10 s window was zero padded to a total length of $2^{14} = 16384$ samples. Changes in spectral power relative to baseline for each electrode contact, time window, and frequency band were represented in decibel scale as follows:

$$P = 10\log_{10}\left(\frac{A}{B}\right)$$

where $B$ is the average power across time in the baseline period, defined here as the 180 s period prior to the onset of bioluminescence, for a given electrode contact and frequency band, and $A$ is the power in a given electrode contact, frequency bin and time bin. In addition to estimating the overall power change in the gamma band we also assessed whether high-power, but brief events in the gamma band may have increased with the introduction of the CTZ. Such events may be less detectable by the multitaper analysis due to the long time windows employed. To detect such short lived gamma events we bandpass filtered the data between 30 and 100 Hz (3rd order Butterworth), Hilbert transformed the data to acquire the analytic signal and took the absolute value to acquire an estimate of the instantaneous amplitude envelope in the gamma band. Next gamma events in the post CTZ period were first defined as any data point that exceeded the 99th percent jackknifed confidence interval of the baseline mean amplitude. An event was then required to exceed this threshold for at least 100 ms (i.e., at least 3 cycles at 30 Hz). Using these criteria, events were summed across the 1200 s post CTZ period for each electrode contact.

To isolate MUA, a bandpass filter (passband: 300 Hz to 3000 Hz, 3rd order Butterworth) was applied to the data. Spikes were defined as data points less than −3 times the standard deviation, where the standard deviation was estimated as the median divided by 0.6745[18]. Spikes were then binned in increments of 1 s to yield a MUA time series for each electrode contact. Each time series was then converted to percent change from baseline.

*Bioluminescence imaging analysis.* For all images, a 3 × 3 pixel median filter was applied to reduce shot noise. For each animal, a circular region with a diameter of 50 pixels was placed in the region directly adjacent to the electrode shank and in front of the surface with the exposed electrode contacts. The mean of these pixels was computed for each image to yield a time series of bioluminescence. Since CTZ was infused into the saline over the craniotomy there was some variability in the onset of bioluminescence across animals. We were specifically interested in the relationship of bioluminescence to changes in gamma-band activity and MUA, so we aligned all data to the onset of bioluminescent signal in the imaging data. We quantified the onset of bioluminescence as the peak of the discreet derivative of the

bioluminescent signal. This method worked well since the bioluminescent signals in these experiments were monotonically increasing with a rapid onset.

*Statistical analysis.* As an initial descriptive statistic we calculated the number of electrode contacts that exceeded baseline for each of the three dependent measures (gamma power (dB) relative to baseline, number of gamma events and MUA percent change from baseline) in the Opsin (+) and Opsin (−) groups. A given electrode contact was considered to have exceeded baseline if its mean value in the period after bioluminescence onset was above the bootstrapped 95th confidence interval of the mean of the baseline period.

For each of the three dependent measures we initially acquired three-dimensional matrices with dimensions electrode contacts, time bins, and animals. In the case of gamma power, the frequency dimension of the spectrograms was collapsed by averaging over the 30-100 Hz frequency bins. To test for significant changes in these measures we took a multi-level approach. First, for each of the dependent measures we pooled all data points across electrode contacts, time bins and animals, keeping the Opsin (+) and Opsin (−) groups separate. These pooled data sets were then submitted to a two-sample Kolmogorov-Smirnov (KS) test to test broadly for differences in the distributions of the Opsin groups. If a given KS test indicated a significant difference between the groups we then averaged across the time dimension and submitted the electrode contacts in each Opsin group to a Mann–Whitney $U$ test. In the case of the gamma measures these tests were performed as one-tailed tests as we hypothesized a priori that upregulation of PV cells in SI via Interluminescence should increase gamma-band activity. The test of MUA differences was performed as a two-tailed test as we had no a priori hypothesis about the directionality of these effects. The KS tests coupled with inspection of the CDFs of the two groups distributions suggested that the Opsin (+) group was positively skewed relative to the Opsin (−) group. Therefore, to sensitively test for group differences at the animal level we computed the 85th percentile value for each of the measures from the array of electrode contacts for each animal. These values were then submitted to Mann–Whitney $U$ tests to test for significant differences between the Opsin group at the animal level. Again, gamma-band activity measures were submitted to one-tailed tests, while the MUA data were submitted to a two-tailed test.

**Reporting summary.** Further information on research design is available in the Nature Research Reporting Summary linked to this article.

## Data availability
All data generated and analysed during this study are included in this published article (and its supplementary information files). All raw data are available from the corresponding authors on reasonable request. The plasmid pAAV-hSyn-hPOMC1-26-sbGLuc-P2A-dTomato is available from Addgene (ID number 176704).

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

## Acknowledgements

The authors thank the members of the Bioluminescence Hub (http://www.bioluminescencehub.org/) for advice and discussions. This research was supported by grants from the US National Institutes of Health (R21MH101525 to U.H.; U01NS099709 to U.H., C.I.M., N.C.S.; R01NS120832 to U.H., C.I.M., N.C.S.), the National Science Foundation (NSF NeuroNex 1707352 to C.I.M., D.L., U.H., N.C.S.), and the W.M. Keck Foundation (to C.I.M., D.L., U.H., J.A.K.). M.P. was a W.M. Keck postdoctoral fellow. The figures were created with BioRender.com.

## Author contributions

M.P. designed and performed in vitro experiments, analyzed data and wrote a draft of the paper. J.M. designed and performed in vivo experiments and analyzed data. R.S., N.F., E.L.C., A.B., A.P. participated in performing experiments. Y.B. participated in data analysis. J.A.K., N.C.S., D.L. provided critical input throughout the studies. C.I.M. and U.H. devised the Interluminescence strategy, R.S. identified the POMC vesicle targeting approach (ref. [11]) used here. U.H. and C.I.M. proposed and directed the overall study. U.H., D.L., and C.I.M. wrote the final paper.

## Competing interests

The authors declare no competing interests.
