## [Transparent Peer Review File · Communications Biology]

Reviewers' comments:

Reviewer #1 (Remarks to the Author):

Prakash and colleagues elegantly combined optogenetics and bioluminescence in synaptically connected neurons as an innovative method to study neuronal circuits. The authors used neurons cultured on multi-electrode arrays to characterize transsynaptic bioluminescence-dependent opsin activation, a new approach they call interluminescence. They nucleofected presynaptic neurons with a small luciferase sbGLuc that is transported into dense core vesicles by adding an N-terminal leader sequence (hPOMC1-26) and the postsynaptic neurons with an excitatory (ChR2(CS)) or inhibitory (hGtACR2) opsin. External light application or CTZ (Luciferin) activated or silenced postsynaptic neurons respectively. Postsynaptic opsin activation required vesicle release (axon cutting or Botox abolished the CTZ effect) and presynaptic activity. The new method seems promising for in vivo circuit analysis and developmental neuroscience, it basically allows to transiently depress or boost an existing connection between two defined brain areas. While the cell culture experiments are well designed and statistically sound, the outcome of the in vivo experiments is surprising and not sufficiently explained. Only one of the two new tools is tested in vivo, the expression pattern is not shown, and the results are highly variable. Otherwise, the manuscript is well written, clearly illustrated and easy to follow.

Major concerns:

1. In vivo experiments: An excitatory step-function opsin is expressed in cortical PV+ interneurons, and multi-unit activity in superficial cortical layers increases after thalamic afferents to interneurons are boosted via interluminescence. This seems paradoxical, as increased activity of interneurons would be expected to decrease pyramidal cell activity. Previously, the Moore lab has shown that light-driven activation of fast-spiking interneurons selectively amplifies gamma oscillations, but in this manuscript, the frequency axis is cut at 15 Hz (Fig. 2d), so we have no information about gamma activity. Please discuss the paradoxical results of interneuron-specific activation by interluminescence (disinhibition?) in the context of published literature about interneuron function. Please depict the circuit (opsin-expressing neurons, pyramidal cells) in Fig. 2 a and f or in a separate schematic.
2. In vivo experiments: Talking about 'opsin' and 'modulation' in this chapter seems to be intentionally vague compared to the in vitro experiments. Please use 'excitatory opsin' or 'ChR2(CS)' for clarity.
3. It would be very attractive to also show data about hGtACR2 effects in vivo, which would be expected to be opposite to activation of the excitatory opsin. This would strengthen the case that both tools work in vivo. In case hGtACR2 is not sufficiently light sensitive to cause interluminescence effects in vivo, this should be clearly stated, too.
4. In vivo experiments: There is no histological analysis shown that might explain the huge experimental variability. Were expression levels (vs electrode position) correlated with effect size in individual animals? Showing that the tools are actually expressed in vivo seems essential.
5. After a single application of CTZ in vivo, for how long is the activity altered?
6. Specificity: The little drawings with the opsin in the spine are suggestive, but it is my understanding that opsin is present on the entire surface of the neuron. It seems likely that mere contact to a luciferase-secreting bouton or dendrite will be sufficient to activate interluminescence currents, even if the two cells are not synaptically connected. I realize that it is not easy to rule out this possibility, but this caveat should at least be mentioned.
7. Please discuss the conditions for dense core vesicle release vs transmitter release. Are single action potentials thought to be sufficient for DCV release? Is DCV release proportional to SV release?
8. Please reformat the manuscript to have 5 main figures and fewer supplemental figures.

Minor

1. Using Gaussia luciferase instead of other enzymes is a smart approach because it still active in low pH plus it is the smallest of the luciferases. This could be better highlighted (e.g. in the discussion).
2. Data from 12 different PV-Cre mice are shown, but the methods say 11 were injected.
3. Naming the method "Interluminescence" creates some grammatical problems. For example (line 133), "Interluminescent activation" sounds very wrong (activation is an abstract noun and therefore not luminescent). It would have to be "activation by interluminescence". Also (Fig. 2, legend), interluminescence configuration instead of "Interluminescent configuration", bioluminescence instead of "bioluminescent light production". Optogenetic actuators instead of "optogenetic sensors".
4. Bioluminescence quantification is called "imaging", but no images are shown anywhere. Please use a more appropriate term and/or describe how the values were extracted from CCD camera images.
5. Fig. 2a: Is 'Luciferin' Coelenterazine (CTZ) in these experiments? Please use consistent abbreviations.

Reviewer #2 (Remarks to the Author):

Optical Synapses: Selective Control of Synaptically-Connected Circuit Elements by Interluminescence

Prakash et al.

In this manuscript, the authors expand upon their previous work¹ to describe an intersectional method for neural circuit manipulation that requires A) pre-synaptic release of luciferase AND B) post-synaptic expression of an opsin (ChR2(C128S)) AND C) local application of a luciferin derivative. In experimental terms, this enables both temporal AND synaptic resolution causal manipulation and is proposed to be useful for systems neuroscience.

To accomplish this, the authors use neuropeptide targeting sequences to package luciferase into (likely) dense core vesicles; this targeting is presumed based on targeting motifs that are added to the luciferase coding sequence. Luciferase is a catalytic peptide that creates a photon de novo as a product of luciferin oxidation. Upon release of luciferase into the synaptic cleft and simultaneous application of a luciferin derivative CTZ, a blue photon is produced. This photon is then absorbed by the post-synaptically-expressed opsin ChR2(CS). This ChR2 variant is a member of the step-function opsin (SFO) family, which stay active for a period of time after photon absorption, enabling them to act as photon integrators and attain maximal ion flux with prolonged exposure to sub-saturating photon densities. The authors further show some data that this same approach can be used with an inhibitory opsin, hGtACR2.

Overall, the approach is interesting in adding to the rich abundance of methodologies for precision manipulation of neural circuitry. The manuscript needs substantial reworking before it could be considered for publication. I would even suggest that it should be revised before being critically evaluated. Despite this, I am happy to provide my critiques in its current format below. These broadly include serious problems with statistics, experimental description, figure labeling, image quality, and discussion.

Major points

1. Figures need more labeling to improve clarity for readers. I am able to decipher them because I am familiar with the molecular components and the approach, but a general reader would be lost. For

example, Figure 1a has a blue cloud that represents blue photons, but this is not labeled. There also isn't any notion that anything has changed with the post-synaptic opsin in its now-presumed-active state. Pre- and post-synaptic neurons are not labeled (although are colored differently). Figure 1b has unlabeled blue triangles above the unlabeled blue bars. I can't read Figure 1c (text is too small and is fuzzy at magnification). There is a data gap in the CTZ conditions in Figs. 1b,c that is not described. Not sure what the unit '0' is on the y-axis of Figs. 1b,c.

a. Similar critiques of Figure 2 (especially 2a,f). Cartoon is not labeled. Would also extend this recommendation to the supplement...For instance, Fig s1a suggests that the luciferase was nucleofected and the opsin was from a transgenic rat (in contrast to figure s2a). etc.

b. Overall, I suggest the authors run the manuscript by someone with a neuroscience background, but not a molecular engineering background, to get their feedback.

2. Why was C128S chosen over opsin variants with longer time constants (e.g. D156A or C128S/D156A)? Or was the C128S/D156A used (as suggested by the methods section 'Virus')? Text needs to be clarified.

3. Would this approach work with non-SFOs (as suggested by results with GtACR)?

4. The only data that suggests that luciferase is indeed packaged into vesicles is the botox experiment (Figure s8). It is much more likely that these are packaged into DCVs than SVs, but the plasmid design (2a separation of luciferase and fluorophore) means that it is not possible to see the distribution of the luciferase itself. It would be helpful to see that luciferase is actually packaged into DCVs, or at least some sort of vesicle, and there is a literature on targeting peptides and small molecules to DCVs, including GFP (see for instance 2,3). If the authors have a non-2a fusion version of their luciferase construct, it would be helpful for the manuscript to see high-resolution imaging of neurons expressing luciferase-XFP in the expected punctate pattern.

5. For Figure 1b, the methods state that only electrodes that showed blue-light modulated activity were used, but it appears that most of the traces in the ladder plot (blue) don't show any change between baseline and light. This needs clarification

a. In addition, the average traces for blue light vs CTZ imply that CTZ is more effective than direct light application. This needs to be clarified; I can't imagine a scenario where luciferase-mediated stimulation is more effective than the direct application of blue light, unless the blue light used was of an incredibly low photon density or very short duration.

6. For Figure 1c, why is the baseline spike rate for the blue light and the CTZ so different (~20-60 vs ~100-300)? Are these data actually from different experiments?

7. There is a paucity of numbers in the manuscript. Figure legends for 1b and 1c describe the data as being from "multiple experiments", but do not give n cells per experiment or total number of experiments. There is also no definition of statistical significance despite there being an asterisk and 'ns' labeled on the panels. None of the averages have error bars or a description of error. The 'Statistics' section of the methods consists, in its entirety of only 'For MEA recordings Mann Whitney test and Wilcoxon matched-pairs signed rank test were used to determine significance.' This issue alone precludes publication.

8. Similar critique of Figure 2: "A significantly greater MUA increase was observed..." with no description of n, numbers, statistical test, or outcome.

9. Why is the luminescence signal peak for Figure 2h-orange shifted nearly 5 minutes later than that of the teal trace? Why is there an increase in activity over time in the control condition in Fig 2g (the orange trace)? And if Figure 2d is describing input from both the VPM and somatosensory nuclei, why isn't there VPM depth activity (the activity shown in Figure 2i) in Figure 2d? There also appears to be data missing around time 0 in Figure 2i.

10. While the authors provide evidence that their approach works in practice, recent controversy regarding biophysical limitations in novel neuroscience methods (e.g magnetogenetics^{4,5}) and hypothesis-driven experimental design dictate that a description of the underlying biophysical principals be included to show theoretical viability. Or, if those do not match up with the empiric data, and explanation should be included.

a. For example, if you need a photon density of X mW over Y seconds in order to drive ChR2(C128S) & whatever that value is for GtACR2, what number of luciferase molecules would you need at an average synaptic cleft distance in order to generate this photon flux with the amount of CTZ that was used? And then, is it reasonable to think that you could have this number of luciferase molecules packaged into dense core vesicles and simultaneously released into the synapse? This is especially important considering that the results shown in this manuscript are comparable to ones presented in previous publications with a fusion approach, where luciferase is physically tethered to the opsin.

b. Considering the small space of the synapse, light power would be expected to decrease from the source to at least the third power, dependent on distance.

11. One of the more important controls that I was happy to see in the paper is the use of synaptic blockers (Figures s5-8). These are critical to show that the observed results are not an artifact derived from poly-synaptic network propagation from a small number of overactive neurons. I am confused by the result in Figure 6b, though – how are the pre-synaptic neurons being stimulated to release luciferase? This is a critical part of the experimental design and needs to be detailed in the figure (or there needs to be an explanation of how this is possible if the neurons were not stimulated). Again, I am concerned about the difference in baseline firing rates in this experiment (Figure 6c).

12. The discussion is sparse, at best. This paper would be able to place this new reagent in context for a wider audience if the following were addressed:

a. What are the limitations of the approach (e.g. can this be used in intact animals?)

b. What is the likely range of opsins this could be used with?

c. How does this tool fit within the broader array of molecule neuroscience tools

Minor points

1. ChR2(CS) first call-out (line 49). This opsin is typically referred to as ChR2(C128S). Change for consistency with the literature.

2. Figure 2c is described before figure 2b in the figure legend.

3. Images of virus expression for Fig 2a,f are missing.

Refs

1. Gomez-Ramirez, M., More, A. I., Friedman, N. G., Hochgeschwender, U. & Moore, C. I. The BioLuminescent-OptoGenetic in vivo response to coelenterazine is proportional, sensitive, and specific in neocortex. *J. Neurosci. Res.* 98, 471–480 (2020).

2. Gubernator, N. G. et al. Fluorescent False Neurotransmitters Visualize Dopamine Release from Individual Presynaptic Terminals. science.sciencemag.org

3. Lang, T. et al. Ca¹⁺-triggered peptide secretion in single cells imaged with green fluorescent protein and evanescent-wave microscopy. *Neuron* 18, 857–863 (1997).

4. Meister, M. Physical limits to magnetogenetics. *arXiv:1604.01359v3 [q-bio.NC]* 5, (2016).

5. Anikeeva, P. & Jasanoff, A. Problems on the back of an envelope. *Elife* 5, (2016).

Reviewer #3 (Remarks to the Author):

The manuscript by Prakash et al. describes a new optogenetic method for controlling synaptically connected circuit elements using bioluminescence. They claim that the method provides the synapse-

selective control of chosen pre- and post-synaptic partners. This method may expand the experimental platform for neuroscience. Although it sounds interesting, the quality of this manuscript is insufficient because of a lack of description of experiments and figures, which prevents the readers, particularly outside the field to easily understand the content.

Major points:

1) "Interluminescence" is a coined word. It's unkind to use it without definition or explanation. This reviewers' understanding of this word is "luminescence produced by luciferase in synaptic cleft". Is this correct? Even if so, however, in the future, names and entities may not always match, as in "optogenetics". The authors should define what the word means.

2) The purpose of using GtACR2 in this study is unclear. Since GtACR2 shows the opposite reaction of ChR2, ON/OFF can be controlled if it is used meaningfully. Although the reaction was confirmed in Fig. 1, it was not used in Fig. 2, etc. It just looks like a control experiment. In addition, there is a lack of explanation for the function of ChR2 and hGtACR2. It is difficult for readers outside the field to understand the difference just by describing "...activated (ChR2(CS)) or inhibited (hGtACR2)..."

3) Supplementary Figure 2 and Figure 3. The results by MEA were shown as a tiling, and the authors described the conclusion as "...co-culture showing strong synchronicity..." and "...synchronous firing activity in both pre ...". First of all, what does each data in the tile mean? Is it just a list of different results? Any association between pre- and post-synapse should be indicated. In addition, there was no quantitative analysis of the synchronicity. If it is mentioned only by the appearance, the manuscript should be modified overall.

4) Supplementary Figure 3. There is no explanation as to which pre-synapse and post-synapse are connected. How did they select electrodes 26, 22, and 17 from many data? An explanation should be given as to why records are missing during CTZ addition. The left group shows that the waveform is greatly disturbed when added CTZ, so it is assumed that it is removed. However, it must not be deleted without a reason. In addition, Sup Fig. 3d shows the enlargement of the results on the left for electrodes 17 and 22, which also removes the data at the time of the addition. This also needs to be mentioned. The stimulation time by CTZ addition appears to be longer than by blue light irradiation. How long does bioluminescence keep for the stimulation? Does it depend on the concentration of CTZ?

5) Supplementary Figure 4 and Figure 7. The release of luciferase from pre-synapse must depend on the transmission of action potential into presynapse, and the amount of luciferase between the synapse varies with the timing. The timing of the 'cut' of synapse connection is also closely related. Therefore, it cannot be easily compared with light irradiation or current stimulation. How much it changes depending on the condition should be discussed. Line79-80, the expression "sufficient luciferase concentration" is very vague.

6) Supplemental Figure 7. The timing of SB addition and stimulation differed among i, ii, iii, and iv. In order to compare each experiment, the authors should stimulate the same conditions and timing.

7) Line144 "Interluminescence provides a unique new platform technology to enable this crucial new level of specificity in circuit control". The technology turns out to be unique in that 'Only postsynaptic cells connected to active presynaptic cells can be selectively controlled'. However, it doesn't offer a vision of how it could be used effectively.

8) Figure2a and f do not clearly show the purpose and the difference between them

Minor points:

1) Some of the characters in the figures are too small to read (Figure1, etc.). In addition, some figures are also too small and too low resolution to understand what they represent (the result of MEA,

Supplementary Figure 2c and 2d, 3, etc.). Figure 2b is not cited in the text.

2) What did the authors use for "vehicle"? Even if it is just a buffer, they must explain.

3) There are many redundant descriptions (Line 87 -94 etc.).

4) Line114 "The pre-synaptic BL component was targeted by injection" does not make sense. Did the authors injected dense core granules containing Gluc?

5) Supplemental Figure 9. The experiments of a and b should be presented further (around Figure1).

6) The authors referred to papers no. 2-6 only because of their luminopsin application. If there is no specific meaning in the citation of each paper, the author should select 2-3 papers.

Response to Reviewers' Critiques

We thank the Reviewers for their thoughtful input: We have addressed all suggestions/comments by adding new data and Figures, and substantially elaborating and revising the text. The new Interluminescence technology described in the accompanying manuscript is, we believe, unique and it has the potential to enable scientists to interrogate and modulate points of cell-cell communication with temporal control. Thanks to the reviewers, the manuscript is now greatly improved.

REV 1

MAJOR

1. In vivo experiments: An excitatory step-function opsin is expressed in cortical PV+ interneurons, and multi-unit activity in superficial cortical layers increases after thalamic afferents to interneurons are boosted via interluminescence. This seems paradoxical, as increased activity of interneurons would be expected to decrease pyramidal cell activity. Previously, the Moore lab has shown that light-driven activation of fast-spiking interneurons selectively amplifies gamma oscillations, but in this manuscript, the frequency axis is cut at 15 Hz (Fig. 2d), so we have no information about gamma activity. Please discuss the paradoxical results of interneuron-specific activation by interluminescence (disinhibition?) in the context of published literature about interneuron function. Please depict the circuit (opsin-expressing neurons, pyramidal cells) in Fig. 2 a and f or in a separate schematic.

Robust Gamma Induction by Interluminescent Drive

We greatly appreciate this comment, and the Reviewer is entirely right: A key entailment of direct increased excitatory thalamic drive of Fast-Spiking Parvalbumin-Positive Interneurons (PV/FS) would be enhanced gamma oscillations.

Following the Reviewer's suggestion, we measured local field potential (LFP) power before and after Interluminescence induction. In agreement with their prediction, and our own prior work (Cardin et al., 2009; Siegle, Pritchett and Moore, 2014) and that of several others, **gamma oscillations showed a robust and selective increase with Interluminescent drive**. This result can be easily appreciated by the mean spectrograms shown in new **Figure 7**, or the changes in gamma power across electrodes, comparing mice expressing the full Interluminescence combination versus those only expressing the presynaptic luciferase. We quantified both mean power increases in gamma and the probability of individual time bins exhibiting high gamma power. As shown in new **Figure 7D**, gamma expression in individual Opsin (+) mice robustly separated them from the Opsin (-) group.

In addition to these effects found in our Initial Cohort, following small thalamic injections, we saw a smaller, more transient gamma increase, as predicted by the time course of bioluminescence production in this cohort. Contamination with 60 Hz line noise required removal of one mouse from this analysis, leaving a relatively small sample (N = 2). Given the smaller effect and sample size, and the confusion that this group appeared to create for some reviewers upon prior submission, we removed it from the presentation.

We now present our MUA findings in the context of these gamma results as, in many contexts, MUA increases are tightly tied to gamma increases. This finding may reflect the order-of-magnitude higher firing rate observed in FS as compared to RS neighbors. We observed, as now

shown in **Figure 7E** and described in the text, an ~2.5-3 fold greater MUA value in the Opsin (+) than Opsin (-) groups. However, these effects were not as consistent across mice as those for gamma induction, and were only significant at the electrode, not the whole mouse, level.

2. In vivo experiments: Talking about 'opsin' and 'modulation' in this chapter seems to be intentionally vague compared to the in vitro experiments. Please use 'excitatory opsin' or 'ChR2(CS)' for clarity.

We appreciate the comment and have now made this change in the text.

3. It would be very attractive to also show data about hGtACR2 effects in vivo, which would be expected to be opposite to activation of the excitatory opsin. This would strengthen the case that both tools work in vivo. In case hGtACR2 is not sufficiently light sensitive to cause interluminescence effects in vivo, this should be clearly stated, too.

We very much agree with the reviewer overall. We demonstrated both postsynaptic excitatory and inhibitory effects of Interluminescence in vitro, then drilled deeper into characterizing key features of Interluminescence focusing on the excitatory opsin. It will definitely be exciting to test a variety of inhibitory opsins in subsequent studies. For those studies hGtACR2 is a good starting point due to its high light sensitivity.

4. In vivo experiments: There is no histological analysis shown that might explain the huge experimental variability. Were expression levels (vs electrode position) correlated with effect size in individual animals? Showing that the tools are actually expressed in vivo seems essential.

The Reviewer's point is well taken, and we included new data to directly address it. While we did not recover anatomy for all mice, we did in a subset, and in a **new figure in the Main Text (Figure 7A) we now show the close (40x) apposition of presynaptic bioluminescent luciferase indexed by red dTomato expressing thalamic axons and neocortical Parvalbumin-positive (fast-spiking) interneurons expressing EYFP tagging step-function opsin expression.** In this image, the canonical 'basket cell' type axonal contact on neighboring neurons is evident. In **Supplemental Figure 7**, we now further show a wide-scale image of our thalamic transduction zone, and neocortical PV expression with robust thalamic neocortical projections.

5. After a single application of CTZ in vivo, for how long is the activity altered?

As shown in the revised **Figure 7**, *in vivo* induction of gamma oscillations sustains for tens of minutes, until we stopped the recording. Importantly, this increase is also true of the bioluminescent signal.

6. Specificity: The little drawings with the opsin in the spine are suggestive, but it is my understanding that opsin is present on the entire surface of the neuron. It seems likely that mere contact to a luciferase-secreting bouton or dendrite will be sufficient to activate interluminescence currents, even if the two cells are not synaptically connected. I realize that it is not easy to rule out this possibility, but this caveat should at least be mentioned.

This point is quite valid, and the reviewer states correctly that opsins can be expected to be expressed across the entire surface of a neuron. Similarly, vesicles containing the luciferase are not necessarily restricted to the synapse but can be found in the soma and along processes. The following considerations support the concept that neuron-to-neuron communication via Interluminescence is restricted to synapses. First, while vesicles (both small and large) are found along the entire neuron, they are preferentially released at presynaptic endings. Second, luciferases are at their highest density right after release into the synaptic cleft. Given that 20 seconds after release addition of luciferin does not elicit a postsynaptic response, it is most likely that luciferases diffused away from the synaptic space do not have a photon density high enough to activate opsins along the neuron. Thus, in contrast to neuropeptide transmission Interluminescence does not seem to have a volumetric effect. We added these considerations in the Discussion section.

7. Please discuss the conditions for dense core vesicle release vs transmitter release. Are single action potentials thought to be sufficient for DCV release? Is DCV release proportional to SV release?

DCV and SV release have different dependencies on firing frequency; LDCV release requires higher frequency stimuli compared to SSV release. Fluorescent labeling of the luciferase and a marker for DCVs revealed colocalization, but a considerable fraction of the luciferases were found in other vesicles. We discuss this observation in the Results and Discussion.

8. Please reformat the manuscript to have 5 main figures and fewer supplemental figures.

We have reformatted from a Brief Communication (2 figures) to a full article (7 figures).

MINOR

1. Using Gaussia luciferase instead of other enzymes is a smart approach because it still active in low pH plus it is the smallest of the luciferases. This could be better highlighted (e.g. in the discussion).

We appreciate the comment. We are highlighting this now in the Results section.

2. Data from 12 different PV-Cre mice are shown, but the methods say 11 were injected.

We apologize for this inconsistency. We have corrected the error in the revised manuscript.

3. Naming the method "Interluminescence" creates some grammatical problems. For example (line 133), "Interluminescent activation" sounds very wrong (activation is an abstract noun and therefore not luminescent). It would have to be "activation by interluminescence". Also (Fig. 2, legend), interluminescence configuration instead of "Interluminescent configuration", bioluminescence instead of "bioluminescent light production". Optogenetic actuators instead of "optogenetic sensors".

We appreciate the opinions of the Reviewer on the naming of our technology. These terms were coined as part of a system of labels we have for BL-OG constructs. The team now has used this system of terms in papers, grants, talks, and multiple trainings (that we provide in using bioluminescence through our NeuroNex Hub). Because we have already introduced this system, and because it has proven great at conveying the understanding we seek in these other contexts—in written and oral format—we are loathe to change them.

4. Bioluminescence quantification is called “imaging”, but no images are shown anywhere. Please use a more appropriate term and/or describe how the values were extracted from CCD camera images.

We apologize for the lack of clarity. Images were recorded from an EMCCD camera, and the traces shown in former **Figure 2** (now **Figure 7 inset**) reflect the average of a circular region of interest (50 pixel diameter) acquired from these images. We have clarified this detail in the *in vivo* Methods and in the Results and legend of **Figure 7**.

5. Fig. 2a: Is ‘Luciferin’ Coelenterazine (CTZ) in these experiments? Please use consistent abbreviations.

Yes, ‘Luciferin’ was meant to refer to Coelenterazine in these experiments. Former **Figure 2a** (Now **Figure 7a**) has been amended for consistency.

REV 2

MAJOR

1. Figures need more labeling to improve clarity for readers. I am able to decipher them because I am familiar with the molecular components and the approach, but a general reader would be lost. For example, Figure 1a has a blue cloud that represents blue photons, but this is not labeled. There also isn’t any notion that anything has changed with the post-synaptic opsin in its now-presumed-active state. Pre- and post-synaptic neurons are not labeled (although are colored differently). Figure 1b has unlabeled blue triangles above the unlabeled blue bars. I can’t read Figure 1c (text is too small and is fuzzy at magnification). There is a data gap in the CTZ conditions in Figs. 1b,c that is not described. Not sure what the unit ‘0’ is on the y-axis of Figs. 1b,c.

All very valuable comments. We were able to address all issues by expanding the previous version of the manuscript from 2 to 7 main figures and by re-drawing all of the schematics.

a. Similar critiques of Figure 2 (especially 2a,f). Cartoon is not labeled. Would also extend this recommendation to the supplement...For instance, Fig s1a suggests that the luciferase was nucleofected and the opsin was from a transgenic rat (in contrast to figure s2a). etc.

We agree with the reviewer that these cartoons were not easily accessible to the reader. We have added labels to the components of the *in vivo* cartoon now in **Figure 7a**.

b. Overall, I suggest the authors run the manuscript by someone with a neuroscience background, but not a molecular engineering background, to get their feedback.

We assume this comment was meant to indicate that the odd, foreshortened format of the prior submission (Brief Communication) was highly confusing, a point that is well-taken: Hopefully, such confusion is no longer a problem in the longer format.

2. Why was C128S chosen over opsin variants with longer time constants (e.g. D156A or C128S/D156A)? Or was the C128S/D156A used (as suggested by the methods section 'Virus')? Text needs to be clarified.

We chose C128S as a starting point for the in vitro experiments based on its higher light sensitivity compared to non-SFOs. For the in vivo experiments we then used the SSFO, ChR2(C128S/D156A), as this was available in the lab as a DIO-AAV. We used hGtACR2 again for its increased light sensitivity compared to other blue light sensing inhibitory opsins. The main consideration was always the higher light sensitivity. We are now working "backwards" (for example, we know from recent experiments that CheRiff also works well).

We clarified the use of the various opsins in the manuscript and the rationale for their choice.

3. Would this approach work with non-SFOs (as suggested by results with GtACR)?

There is no theoretical reason that it would not work (as mentioned above, we know that non-SFOs work well, in addition to hGtACR2). We had to start somewhere for the initial studies and will follow up with a more systematic study comparing opsins with different features and light sensitivities in the future.

4. The only data that suggests that luciferase is indeed packaged into vesicles is the botox experiment (Figure s8). It is much more likely that these are packaged into DCVs than SVs, but the plasmid design (2a separation of luciferase and fluorophore) means that it is not possible to see the distribution of the luciferase itself. It would be helpful to see that luciferase is actually packaged into DCVs, or at least some sort of vesicle, and there is a literature on targeting peptides and small molecules to DCVs, including GFP (see for instance 2,3). If the authors have a non-2a fusion version of their luciferase construct, it would be helpful for the manuscript to see high-resolution imaging of neurons expressing luciferase-XFP in the expected punctate pattern.

We generated a POMC-sbGLuc-eGFP fusion protein and collected images from neurons expressing this construct. The punctate pattern is consistent with the luciferases located in vesicles and staining with a marker for DCVs revealed co-localization to some extent. This information has been added as Supplementary Figure 6 and to the Results section.

5. For Figure 1b, the methods state that only electrodes that showed blue-light modulated activity were used, but it appears that most of the traces in the ladder plot (blue) don't show any change between baseline and light. This needs clarification.

a. In addition, the average traces for blue light vs CTZ imply that CTZ is more effective than direct light application. This needs to be clarified; I can't imagine a scenario where luciferase-mediated

stimulation is more effective than the direct application of blue light, unless the blue light used was of an incredibly low photon density or very short duration.

We now provide the Figures as high quality pdf files, allowing the Reviewer to zoom in and revealing that all electrodes included in the analysis show an increase (excitatory opsin) or decrease (inhibitory opsin) in spiking.

These are recordings from neural populations grown on multi electrode arrays, not from individual neurons. Spiking activity varies between electrodes: Therefore, we display the data in ladder plots. For each individual electrode it shows the spiking activity before the treatment and after, reflecting the variance in 'starting activity' between electrodes and recordings.

For all experiments we assess the difference between starting point (before) and endpoint (after) with the sign of the spiking change as the critical parameter. The absolute number of spikes in this design does not tell much about the magnitude of the neural response, and thus do not allow us to compare LED activation to bioluminescence activation.

6. For Figure 1c, why is the baseline spike rate for the blue light and the CTZ so different (~20-60 vs ~100-300)? Are these data actually from different experiments?

We previously counted numbers of spikes across different time periods to also capture the average length of the responses (LED 10 seconds, CTZ 100 seconds). We re-calculated and re-graphed all recordings to display 5 seconds before/after consistently across all recordings.

7. There is a paucity of numbers in the manuscript. Figure legends for 1b and 1c describe the data as being from "multiple experiments", but do not give n cells per experiment or total number of experiments. There is also no definition of statistical significance despite there being an asterisk and 'ns' labeled on the panels. None of the averages have error bars or a description of error. The 'Statistics' section of the methods consists, in its entirety of only 'For MEA recordings Mann Whitney test and Wilcoxon matched-pairs signed rank test were used to determine significance.' This issue alone precludes publication.

We now specify number of experiments and number of electrodes recorded in the Results section and the Figure legends. Statistical significance is defined in each Figure legend or associated text. The averages in the ladder plots are indicated by thicker lines – in addition to lines for each individual recorded electrode. Together, these show the average and the spread of all data.

The Statistics section for the in vitro experiments has been substantially expanded.

8. Similar critique of Figure 2: "A significantly greater MUA increase was observed..." with no description of n, numbers, statistical test, or outcome.

We now provide extensive quantification of our gamma and MUA analyses in the *in vivo* Results.

9. Why is the luminescence signal peak for Figure 2h-orange shifted nearly 5 minutes later than that of the teal trace? Why is there an increase in activity over time in the control condition in Fig 2g (the orange trace)? And if Figure 2d is describing input from both the VPM and somatosensory

nuclei, why isn't there VPM depth activity (the activity shown in Figure 2i) in Figure 2d? There also appears to be data missing around time 0 in Figure 2i.

These data are no longer reported in the document, due to the evident confusion they caused. That said, the difference in timing in the traces in former Figure 2h are likely due to the method of CTZ delivery we employed. That is, CTZ was infused into saline sitting over the exposed brain. In some cases, the CTZ diffused throughout the saline and rapidly found luciferase whereas in other cases CTZ took a little longer to find its target. Every attempt was made to place the infusion tubing at a consistent distance from the opening of the craniotomy.

10. While the authors provide evidence that their approach works in practice, recent controversy regarding biophysical limitations in novel neuroscience methods (e.g magnetogenetics^{4,5}) and hypothesis-driven experimental design dictate that a description of the underlying biophysical principals be included to show theoretical viability. Or, if those do not match up with the empiric data, and explanation should be included.

a. For example, if you need a photon density of X mW over Y seconds in order to drive ChR2(C128S) & whatever that value is for GtACR2, what number of luciferase molecules would you need at an average synaptic cleft distance in order to generate this photon flux with the amount of CTZ that was used? And then, is it reasonable to think that you could have this number of luciferase molecules packaged into dense core vesicles and simultaneously released into the synapse? This is especially important considering that the results shown in this manuscript are comparable to ones presented in previous publications with a fusion approach, where luciferase is physically tethered to the opsin.

We very much agree that biophysical calculations can be helpful to estimate what is possible. For ChR2(H134R) such calculations have been presented (Hegemann 2011; Lin 2011). In our experiments the photon density needed for ChR2(C128S) or hGtACR2 is not known but can be assumed to be considerably less than that needed for ChR2(H134R) based on their higher light sensitivity. The quantum yield of sbGLuc is not known but can be estimated to be higher than that of luciferases for which this number is known (firefly, Nanoluc). While calculations of possible light emission from a given number of luciferase molecules can be determined, the number released, how many vesicles are fused upon activation, how many presynaptic axons contact a given recipient cell, and the exact concentration of luciferin in the synapse are all unknown for any even semi-realistic use context. As such, almost any guesstimate made by filling in value for all of these variables (and others, such as relative pH in the synapse, etc) do not provide a clear path to the reality of an actual biological context and would not be the basis for either doing the experiment or not doing the experiment.

Therefore, we have approached bioluminescence activation of opsins (BL-OG) from an empirical starting point. BL-OG using wildtype *Gaussia* luciferase tethered to ChR2(H134R) results in a very small depolarization; An opsin with higher light sensitivity (VChR1) results in a larger depolarization (Berglund 2013); and, using this opsin with a luciferase that provides ~10x higher photon emission (sbGLuc) results in action potential firing (Berglund 2016). Accordingly, when separating light emitter and light sensor for Interluminescence, we started with the brightest version of *Gaussia* luciferase, sbGLuc, and opsins with thousand-fold higher light sensitivities than VChR1 (ChR2(C128S), hGtACR2). Our experiments provide a practical starting point for trying alternative combinations of luciferases, opsins, and luciferins in different sets of neurons.

b. Considering the small space of the synapse, light power would be expected to decrease from the source to at least the third power, dependent on distance.

As the synaptic cleft is closer than the wavelength of the light, we assume there is neither absorption nor scattering. Further, light emission caused by the luciferase encountering the substrate CTZ could happen at the time the luciferase leaves the fused vesicle and enters the synaptic cleft, or after the luciferase has travelled across the synaptic cleft, reached the opsin and at that point encounters CTZ.

11. One of the more important controls that I was happy to see in the paper is the use of synaptic blockers (Figures s5-8). These are critical to show that the observed results are not an artifact derived from poly-synaptic network propagation from a small number of overactive neurons. I am confused by the result in Figure 6b, though – how are the pre-synaptic neurons being stimulated to release luciferase? This is a critical part of the experimental design and needs to be detailed in the figure (or there needs to be an explanation of how this is possible if the neurons were not stimulated). Again, I am concerned about the difference in baseline firing rates in this experiment (Figure 6c).

After addition of blockers the spontaneous neural population activity stops (new **Fig. 4 b(i)**; most likely due to disruption of synaptically coupled network activity within cortical luciferase expressing and striatal opsin expressing neurons). If CTZ is administered together with or immediately after addition of blockers, enough luciferase is available, from the spontaneous activity before addition of blockers (old Fig. 6b - new **Fig. 4b(ii)**). This situation is not the case when CTZ administration is delayed by 20 seconds or more (new **Fig.4d(i)**); however, even 20 seconds after addition of blockers CTZ administration can trigger neural firing provided that acute electrical stimulation is applied immediately preceding CTZ application (new **Fig.4d(iii)**).

12. The discussion is sparse, at best. This paper would be able to place this new reagent in context for a wider audience if the following were addressed:

- a. What are the limitations of the approach (e.g. can this be used in intact animals?)*
- b. What is the likely range of opsins this could be used with?*
- c. How does this tool fit within the broader array of molecule neuroscience tools*

With the reorganization of the manuscript from Brief Communication to full article we now have a more extensive Discussion section that incorporates the reviewer's suggestions.

Minor points

1. ChR2(CS) first call-out (line 49). This opsin is typically referred to as ChR2(C128S). Change for consistency with the literature.

Done

2. Figure 2c is described before figure 2b in the figure legend.

Figure 2 is now **Figure 7** and we have corrected the legend to reflect the ordering of the figures.

3. Images of virus expression for Fig 2a,f are missing.

The Reviewer's point is well taken, and we included new data to directly address it. In a **new figure in the Main Text (Figure 7A) we now show the close (40x magnification) apposition of presynaptic bioluminescent luciferase indexed by red dTomato expressing thalamic axons in neocortex, and neocortical Parvalbumin-positive (fast-spiking) interneurons expressing ChR2-EYFP**. In this image, the canonical 'basket cell' type axonal contact on neighboring neurons is evident. In **Supplemental Information**, we now further show a wide-scale image of our thalamic transduction zone, and of neocortical PV expression with robust thalamic neocortical projections.

Refs

1. Gomez-Ramirez, M., More, A. I., Friedman, N. G., Hochgeschwender, U. & Moore, C. I. The BioLuminescent-OptoGenetic in vivo response to coelenterazine is proportional, sensitive, and specific in neocortex. *J. Neurosci. Res.* 98, 471–480 (2020).
2. Gubernator, N. G. et al. Fluorescent False Neurotransmitters Visualize Dopamine Release from Individual Presynaptic Terminals. *science.sciencemag.org*
3. Lang, T. et al. Ca¹⁺-triggered peptide secretion in single cells imaged with green fluorescent protein and evanescent-wave microscopy. *Neuron* 18, 857–863 (1997).
4. Meister, M. Physical limits to magnetogenetics. *arXiv:1604.01359v3 [q-bio.NC]* 5, (2016).
5. Anikeeva, P. & Jasanoff, A. Problems on the back of an envelope. *Elife* 5, (2016).

Reviewer #3 (Remarks to the Author):

The manuscript by Prakash et al. describes a new optogenetic method for controlling synaptically connected circuit elements using bioluminescence. They claim that the method provides the synapse-selective control of chosen pre- and post-synaptic partners. This method may expand the experimental platform for neuroscience. Although it sounds interesting, the quality of this manuscript is insufficient because of a lack of description of experiments and figures, which prevents the readers, particularly outside the field to easily understand the content.

Major points:

1) "Interluminescence" is a coined word. It's unkind to use it without definition or explanation. This reviewers' understanding of this word is "luminescence produced by luciferase in synaptic cleft". Is this correct? Even if so, however, in the future, names and entities may not always match, as in "optogenetics". The authors should define what the word means.

We hope to have addressed the reviewer's very valid point in the Abstract: Luciferase-generated light, originating from a presynaptic axon terminal, modulates an opsin in its postsynaptic target. Vesicular-localized luciferase is released into the synaptic cleft in response to presynaptic activity, creating a real-time 'Optical Synapse'. ... We validate synaptic 'Interluminescence' by multi-electrode recording in cultured neurons and in mice *in vivo*.

2) The purpose of using GtACR2 in this study is unclear. Since GtACR2 shows the opposite reaction of ChR2, ON/OFF can be controlled if it is used meaningfully. Although the reaction was confirmed in Fig. 1, it was not used in Fig. 2, etc. It just looks like a control experiment. In addition, there is a lack of explanation for the function of ChR2 and hGtACR2. It is difficult for readers outside the field to understand the difference just by describing "...activated (ChR2(CS)) or inhibited (hGtACR2)..."

In the revised paper we hope we clarified the use of different opsins in different parts of the study. We introduce Interluminescence starting with the general principle, i.e. presynaptic release of luciferase and postsynaptic activation of an optogenetic element (Fig. 2a). The consequences for the postsynaptic neuron are determined by the biophysical characteristics of the opsin: depolarization if the opsin is a cation channel, and hyperpolarization if the opsin is an anion channel. In the rest of Fig. 2 we then demonstrate both applications: activation by using the excitatory opsin ChR2(C128S) and inhibition by using the inhibitory opsin hGtACR2 expressed in postsynaptic partners, respectively (Fig. 2 c and d).

In the Results section we expanded on the description of this experiment: “the excitatory step function opsin ChR2(C128S) and the inhibitory anion channel hGtACR2”; “We used blue light to activate opsins directly and showed that this increased (ChR2(C128S)) or decreased (hGtACR2) the activity of the culture as expected based on the type of opsin expressed. We then added the luciferin Coelenterazine (CTZ), the substrate for Gaussia luciferases and observed increased (ChR2(C128S)) or decreased (hGtACR2) spontaneous activity consistent with the expressed opsin.”

3) Supplementary Figure 2 and Figure 3. The results by MEA were shown as a tiling, and the authors described the conclusion as “...co-culture showing strong synchronicity...” and “...synchronous firing activity in both pre ...”. First of all, what does each data in the tile mean? Is it just a list of different results? Any association between pre- and post-synapse should be indicated. In addition, there was no quantitative analysis of the synchronicity. If it is mentioned only by the appearance, the manuscript should be modified overall.

Our apologies for the lack of clarity. **Supplementary Figure 2 panel a(i)** shows neural spiking for each of the 60 electrodes of the multi electrode array over 300 sec. Reflecting the initial plating of neurons in two separate chambers, activity can be detected for the upper and lower three rows of electrodes, but not in the two rows in the middle that separate cortical (upper) from hippocampal (lower) neurons. This gap is covered by neural processes, mainly from the cortical neurons to the striatal neurons. In **panel a(ii)** activity is shown as a raster plot, with the individual electrodes aligned from top to bottom on the Y axis. The time scale (X axis) is zoomed in to show the rhythmic firing pattern and the synchronization between upper (cortical) and lower (hippocampal) neurons. **Panel a(iii)** shows representative recordings of a pair of pre (upper cortical)- and post (lower hippocampal)-synaptic electrodes, revealing a delay in start of the firing event of the post-synaptic electrode, characteristic of synaptic transmission. This delay in postsynaptic firing can also be seen in a raster plot from the entire co-culture in **panel a(iv)**. The synchronicity of co-culture firing is apparent from the displays of spiking and the raster plots and is presented to confirm that the pre- and postsynaptic neurons are synaptically connected.

In contrast to the un-cut co-culture presented in panel a, the analogous screenshots and raster plots in **panel b** display the recordings after the connections between pre- (cortical) and post-(hippocampal) neurons have been severed. While synchronicity within each population is preserved, it is lost between the cortical and hippocampal populations.

Similarly, **Supplementary Figure 3** shows another example of a MEA recording from a co-culture of cortical luciferase expressing presynaptic neurons and hippocampal opsin expressing postsynaptic neurons. The apparent synchronicity demonstrates that the two neural populations are synaptically connected. The subsequent **panels b - e** show the reaction of the two populations to various stimuli.

4) Supplementary Figure 3. There is no explanation as to which pre-synapse and post-synapse are connected. How did they select electrodes 26, 22, and 17 from many data? An explanation should be given as to why records are missing during CTZ addition. The left group shows that the waveform is greatly disturbed when added CTZ, so it is assumed that it is removed. However, it must not be deleted without a reason. In addition, Sup Fig. 3d shows the enlargement of the results on the left for electrodes 17 and 22, which also removes the data at the time of the addition. This also needs to be mentioned. The stimulation time by CTZ addition appears to be longer than by blue light irradiation. How long does bioluminescence keep for the stimulation? Does it depend on the concentration of CTZ?

These are co-cultures of cortical luciferase-expressing neurons and hippocampal opsin-expressing neurons, with many cortical neurons synaptically connected with hippocampal neurons. Further, as these are recordings from multi electrode arrays, spiking reflects activity of neural populations located on or near a given electrode. Electrical stimulation of 5 individual electrodes among the cortical neuron population increases firing in all electrodes, both the cortical and hippocampal population (**panel c**). In contrast, blue LED exposure of the culture increases firing exclusively in the opsin expressing hippocampal neurons (as expected; **panel d**). Exposure of the culture to CTZ again increases firing exclusively in the opsin expressing hippocampal neurons (**panel e**), in this case driven by light emitted from luciferases released from presynaptic cortical neurons.

The zoomed-in displays of individual electrodes are representative examples.

The “white-out” immediately after addition of CTZ blocks out the artifact introduced by addition of liquid to the culture. We now indicate this in each figure legend.

Blue light emission from the luciferase after addition of CTZ usually lasts longer than 10 seconds (the time blue LED is on). The length of the effect depends on the concentration of CTZ as well as on the starting activity of the neuronal population.

5) Supplementary Figure 4 and Figure 7. The release of luciferase from pre-synapse must depend on the transmission of action potential into presynapse, and the amount of luciferase between the synapse varies with the timing. The timing of the ‘cut’ of synapse connection is also closely related. Therefore, it cannot be easily compared with light irradiation or current stimulation. How much it changes depending on the condition should be discussed. Line79-80, the expression “sufficient luciferase concentration” is very vague.

We added more explanation to both text and figures legends to clarify the design of the experiments and their interpretation (new **Figures 3 and 4**).

6) Supplemental Figure 7. The timing of SB addition and stimulation differed among i, ii, iii, and iv. In order to compare each experiment, the authors should stimulate the same conditions and timing.

Varying the time between SB addition and electrical stimulation goes to the core of this experiment. We apologize for the lack of clarity and have added more explanation to both text and figures legends (new **Figure 4**).

7) Line 144 *“Interluminescence provides a unique new platform technology to enable this crucial new level of specificity in circuit control”*. The technology turns out to be unique in that *‘Only postsynaptic cells connected to active presynaptic cells can be selectively controlled’*. However, *it doesn’t offer a vision of how it could be used effectively*.

We have added to the Discussion section addressing the reviewer’s very valid point.

8) *Figure 2a and f do not clearly show the purpose and the difference between them*

The Reviewer’s point is well taken: Because this contrast proved confusing for readers, whereas the robust main effects of Interluminescence-driven gamma increases are easy to appreciate, we have removed the smaller thalamic injections from the document.

Minor points:

1) *Some of the characters in the figures are too small to read (Figure 1, etc.). In addition, some figures are also too small and too low resolution to understand what they represent (the result of MEA, Supplementary Figure 2c and 2d, 3, etc.). Figure 2b is not cited in the text.*

We now submit the figures as TIF files with better resolution. Further, we revised all figures, together with their legends and description in the main text.

2) *What did the authors use for “vehicle”? Even if it is just a buffer, they must explain.*

We now define ‘vehicle’ in the main text, the figure legends, and the Methods section.

3) *There are many redundant descriptions (Line 87 -94 etc.).*

With the reorganization of the manuscript from a Brief Communication to a full article we have reworked the entire text.

4) Line 114 *“The pre-synaptic BL component was targeted by injection” does not make sense. Did the authors inject dense core granules containing Gluc?*

We appreciate the need for clarity, and we have now edited the text accordingly.

5) *Supplemental Figure 9. The experiments of a and b should be presented further (around Figure 1).*

With the reorganization of the manuscript this previous Supplementary figure is now one of the main figures (Fig. 6).

6) *The authors referred to papers no. 2-6 only because of their luminopsin application. If there is no specific meaning in the citation of each paper, the author should select 2-3 papers.*

With the reorganization of the manuscript the references were re-worked as well.

Reviewers' comments:

Reviewer #1 (Remarks to the Author):

The authors have substantially expanded the manuscript, adding new data and improving the discussion and the methods section. I am not sure how the large landscape figures will look like in print, but I leave the layout questions to the journal. The 'during behavior' claim in the abstract should be removed - in vivo experiments were under anesthesia and terminal. Otherwise, my concerns have been addressed, and I have no further questions.

Reviewer #2 (Remarks to the Author):

Prakash et al. here submit a revised manuscript whereby they develop and validate a novel genetic/optical approach that enables temporally-defined manipulation of targeted pre-/post-synaptic neural circuit elements. They accomplish this by expressing a vesicle-targeted form of luciferase in the pre-synaptic population and optically-modulated ion channels (optogenetic tools) in the post-synaptic population.

The manuscript has been substantially re-worked since the prior draft and the improved version has largely addressed my concerns and does an effective job of telling the story.

The major concerns I raised have been addressed as follows:

1. The text and figures are improved, consistent, and will be accessible to a general neuroscience audience. Cartoons provide excellent context to reader.
2. Additional experiments have been conducted to assay vesicle distribution of the luciferase construct. In addition, the likely inclusion of the luciferase actuator in DCVs is addressed in the results and discussion.
3. Description of experimental details and statistics are improved from the original version.
4. I appreciate the thoughtful response of the authors to my request for back-of-the-envelope calculations to validate that their observed results are compatible with what is possible in theory. Although they declined to complete such a calculation, they have added a paragraph to the discussion that touches on this question and characterize their approach a step in the progression of development of this line of tools.
5. The expanded discussion adds a lot to the manuscript and will help readers place this tool among others.

Most of the controls are done well - missing sometimes is a CTZ alone control (for instance, Fig 2 c,d). However, the question of whether CTZ alone is able to modulate post-synaptic activity is assayed in Figure 3, in the 'cut' luciferase condition, which is sufficient.

I had considered another issue during the intervening period -- namely that it may be possible for luciferase that is housed within primed vesicles at the synapse (where luciferase in synaptic vesicles is close to the synapse but has not been released) to contribute to post-synaptic activating independent of vesicle release. I believe that this is reasonably addressed with two sets of experiments: 1) the BoNT work (prevents vesicle release through degradation of vesicle fusion machinery) and 2) the cut axon component of the co-culture work (which leaves the terminals intact, but without action-potential driven vesicle fusion. If luciferase within the primed/docked vesicles contributes to post-synaptic opsin modulation, then it appears minimal. It may be worth the authors time to mention this somewhere in the discussion, if they find it useful, as readers may wonder about this possibility.

Minor comments:

1. In Interluminescence modulates spontaneous neural activity in culture.. First sentence, "...to synaptic vesicles in..." As noted by authors later, they target luciferase to both DCVs and SSVs.

Consider making more general to encompass both targeted vesicle types.

2. In Interluminescence-mediated activation of postsynaptic neurons requires opsins. Second to last sentence "In cultures expressing inactive ChR2(C128S)..." unclear if referring here to the C128S/E97R/D253A mutant or to C128S that has been inactivated by yellow light.

3. In Discussion Last part of paragraph 3 and first part of paragraph 4 are redundant.

4. Figure 4:

a. Please identify opsin used in experiment somewhere.

b. Statistical comparisons in panels c and e are inconsistent. Are comparisons not shown ns? For instance, in panel e, there is a single comparison shown between after values for CTZ added immediately and vehicle added immediately, but no other similar comparisons...

5. Figure 5:

a. Please identify opsins used in experiment somewhere.

6. Figure 7:

a. I like a; could be even more clear if you showed where you injected which virus in the anatomy cartoon.

b. Insets for b are difficult to read; move to separate panel if possible.

7. Figure S1:

a. Please add time scale bar

b. Not sure what 10 um means for CTZ...10 ul? uM?

8. Figure S3 & S4:

a. Please indicate which opsin is used for these experiments somewhere in figure or legend.

Reviewer #3 (Remarks to the Author):

The authors have nicely addressed the reviewers' comments. I have just some minor concerns as follows.

1. It is not confirmed, how the close proximity of luciferases interrupts the synaptic cleft, however the expression of opsin closely affects total bioluminescence transmission. How much close proximity would influence the synaptic activation?

2. The mixing of interluminescence and interluminescent seems not to sound for the general reader. They should carefully treat the word inappropriate expression. For example, the authors said, the precise photon density required to create the Interluminescent effects, however, this could be interluminescence effects.

3. In the following sentence, "the luciferin Coelenterazine (CTZ), the substrate for Gaussia luciferases", there is a redundancy of luciferin and substrate. Therefore, they should use either luciferin or substrate.

4. Why interluminescence doesn't have a volumetric effect than orthogonal neuropeptide? Is there any evidence of their claim?

5. In Figure 7, the inside letters are too small. They should enlarge the font sizes.

Response to Reviewers' Critiques

We thank the Reviewers for their careful reading of our revised manuscript and the excellent comments. We believe we have addressed all remaining suggestions/comments in both text and Figures.

Reviewers' comments:

Reviewer #1 (Remarks to the Author):

The authors have substantially expanded the manuscript, adding new data and improving the discussion and the methods section. I am not sure how the large landscape figures will look like in print, but I leave the layout questions to the journal. The 'during behavior' claim in the abstract should be removed - in vivo experiments were under anesthesia and terminal. Otherwise, my concerns have been addressed, and I have no further questions.

We deleted "during behavior":

Abstract

Understanding percepts, engrams and actions requires methods for selectively modulating synaptic communication between specific subsets of interconnected cells. Here, we develop an approach to control synaptically connected elements using bioluminescent light: Luciferase-generated light, originating from a presynaptic axon terminal, modulates an opsin in its postsynaptic target. Vesicular-localized luciferase is released into the synaptic cleft in response to presynaptic activity, creating a real-time 'Optical Synapse'. Light production is under experimenter-control by introduction of the small molecule luciferin. Signal transmission across this optical synapse is temporally defined by the presence of both the luciferin and presynaptic activity. We validate synaptic 'Interluminescence' by multi-electrode recording in cultured neurons and in mice *in vivo*. Interluminescence represents a powerful approach to achieve synapse-specific and activity-dependent circuit control ~~during behavior~~ *in vivo*.

Reviewer #2 (Remarks to the Author):

Prakash et al. here submit a revised manuscript whereby they develop and validate a novel genetic/optical approach that enables temporally-defined manipulation of targeted pre-/post-synaptic neural circuit elements. They accomplish this by expressing a vesicle-targeted form of luciferase in the pre-synaptic population and optically-modulated ion channels (optogenetic tools) in the post-synaptic population.

The manuscript has been substantially re-worked since the prior draft and the improved version has largely addressed my concerns and does an effective job of telling the story.

The major concerns I raised have been addressed as follows:

- 1. The text and figures are improved, consistent, and will be accessible to a general neuroscience audience. Cartoons provide excellent context to reader.*
- 2. Additional experiments have been conducted to assay vesicle distribution of the luciferase construct. In addition, the likely inclusion of the luciferase actuator in DCVs is addressed in the results and discussion.*
- 3. Description of experimental details and statistics are improved from the original version.*
- 4. I appreciate the thoughtful response of the authors to my request for back-of-the-envelope calculations to validate that their observed results are compatible with what is possible in theory. Although they declined to complete such a calculation, they have added a paragraph to the*

discussion that touches on this question and characterize their approach a step in the progression of development of this line of tools.

5. The expanded discussion adds a lot to the manuscript and will help readers place this tool among others.

Most of the controls are done well - missing sometimes is a CTZ alone control (for instance, Fig 2 c,d). However, the question of whether CTZ alone is able to modulate post-synaptic activity is assayed in Figure 3, in the 'cut' luciferase condition, which is sufficient.

I had considered another issue during the intervening period -- namely that it may be possible for luciferase that is housed within primed vesicles at the synapse (where luciferase in synaptic vesicles is close to the synapse but has not been released) to contribute to post-synaptic activating independent of vesicle release. I believe that this is reasonably addressed with two sets of experiments: 1) the BoNT work (prevents vesicle release through degradation of vesicle fusion machinery) and 2) the cut axon component of the co-culture work (which leaves the terminals intact, but without action-potential driven vesicle fusion. If luciferase within the primed/docked vesicles contributes to post-synaptic opsin modulation, then it appears minimal. It may be worth the authors time to mention this somewhere in the discussion, if they find it useful, as readers may wonder about this possibility.

We appreciate the reviewer's thoughtful comment. We intend to address this and other questions about Interluminescence in much more detail than we were able to do here in upcoming studies.

Minor comments:

1. In Interluminescence modulates spontaneous neural activity in culture.. First sentence, "...to synaptic vesicles in..." As noted by authors later, they target luciferase to both DCVs and SSVs. Consider making more general to encompass both targeted vesicle types.

We kept it more general at this point by removing the specifier "synaptic":

Interluminescence modulates spontaneous neural activity in culture. We targeted the blue light emitting luciferase sbGLuc, a bright Gaussia luciferase variant¹⁰, to ~~synaptic~~-vesicles in cortical neurons using the vesicle targeting sequence of the human pro-opiomelanocortin pro-peptide (hPOMC1-26)^{11,12}. The targeting construct also contained the reporter gene dTomato attached to sbGLuc via a P2A cleavage sequence (**Fig. 2a**). In addition to being a bright photon source, sbGLuc is favorable because it is also stable at the lower pH levels of synaptic vesicles^{13,14}.

2. In Interluminescence-mediated activation of postsynaptic neurons requires opsins. Second to last sentence "In cultures expressing inactive ChR2(C128S)..." unclear if referring here to the C128S/E97R/D253A mutant or to C128S that has been inactivated by yellow light.

We clarified this by noting the full name of the construct:

signed rank test). Second, we used a non-functional opsin mutant ChR2(C128S)-E97R-D253A that does not produce photocurrent²⁰ (**Fig. 6d** schematic, **Fig. 6e** upper trace). In cultures expressing inactive ChR2(C128S)-E97R-D253A in postsynaptic neurons, CTZ generated bioluminescence, but no increase in spiking (**Fig. 6e**, lower trace, **6f** ladder plot; SB alone (after), n=21, v/s SB + CTZ (after), n= 49, p=0.7870; Mann-Whitney test). These data indicate that Interluminescence is mediated by photocurrent generation following bioluminescent activation of the opsin.

3. In Discussion Last part of paragraph 3 and first part of paragraph 4 are redundant.

We removed the duplicated text:

presynaptic vesicle fusion. BoNT inhibits both small SSVs and large LDCVs³⁷ and immunohistochemistry revealed colocalization of luciferases in both dopamine β -hydroxylase-containing and non-dopamine β -hydroxylase-containing vesicles¹⁹. Thus, although we used the POMC sorting signal to concentrate luciferase in peptide-containing LDVs, our results suggest that luciferase was present in LDCVs as well as non-peptide containing synaptic vesicles.

~~BoNT inhibits both small SSVs and large LDCVs³⁷, and immunohistochemistry revealed colocalization of luciferases in dopamine β -hydroxylase-containing and non-dopamine β -hydroxylase-containing vesicles¹⁹. Achieving a higher degree of specific targeting of luciferases to specific vesicles, either LDCVs or SSVs could provide a way to establish functional connectivity~~

4. Figure 4:

a. Please identify opsin used in experiment somewhere.

The opsin now is being identified in the figure legend.

b. Statistical comparisons in panels c and e are inconsistent. Are comparisons not shown ns? For instance, in panel e, there is a single comparison shown between after values for CTZ added immediately and vehicle added immediately, but no other similar comparisons...

Panels c and e are now consistent.

Fig. 4. Interluminescence elicits postsynaptic firing increase in the presence of synaptic blockers dependent on presynaptic neuronal activity. a Illustrations showing release of synaptic vesicle contents (neurotransmitters: yellow spheres, luciferases: blue enzymes) with spontaneous presynaptic activity inducing postsynaptic responses with transmitters alone (left panel), with transmitters and bioluminescent activation of opsins in the presence of CTZ (middle panel), and the effect of application of synaptic blockers (SB), allowing to isolate the effects of bioluminescence-mediated synaptic transmission (right panel). b Traces from representative electrodes of opsin (CHR2(C128S)) expressing population applying to the culture (i) synaptic blockers alone, (ii) synaptic blockers together with CTZ, or (iii) synaptic blockers together with vehicle. c Ladder plots of recordings under the conditions depicted in (b) from electrodes across opsin expressing populations comparing number of spikes 5 seconds before and after (i) synaptic blockers alone, (ii) synaptic blockers together with CTZ, or (iii) synaptic blockers together with vehicle. (i) SB alone, n=27, (ii) SB + CTZ, n=37, (iii) SB + vehicle, n=38; SB alone (after) v/s SB + CTZ (after), p<0.0001; SB + CTZ (after) v/s SB + vehicle (after), p<0.0001; SB alone (after) v/s SB + vehicle (after), P=0.7305; Mann-Whitney test. d Traces from representative electrode recordings of opsin expressing population applying to the culture synaptic blockers followed after ~20 seconds by application of (i) CTZ, (ii) electrical stimulation, and electrical stimulation together with either (iii) CTZ or (iv) vehicle. e Ladder plots of recordings under the conditions depicted in (d) across populations. (i), n=18, SB (after) v/s CTZ added ~20s later (after), p=0.4022; (ii), n=35, SB (after) v/s electrical stimulation (after) p>0.9999; (iii), n= 10; electrical stimulation (after) v/s immediate CTZ treatment (after), p<0.0001; (iv), n=35, electrical stimulation (after) v/s immediate vehicle treatment (after), p=0.8553; Mann-Whitney test. **Significant increase in activity of opsin expressing populations is observed only when CTZ is applied immediately following electrical stimulation and not when CTZ or electrical stimulation are applied by themselves nor when vehicle is applied immediately following electrical stimulation (yellow bars: immediate CTZ (after), (n=10) v/s CTZ added ~20s later (after), (n=18); electrical stimulation (after), (n=35); immediate vehicle (after), (n=35); p<0.0001; Mann-Whitney test.** The artifacts due to addition of reagents in MEAs are overlaid by a vertical white bar in the recording traces (the white gap right after addition of SB, CTZ or vehicle). Artifacts due to electrical stimulation are visible under the red bolts. ns, not significant; ****, p<0.0001

5. Figure 5:

a. Please identify opsin used in experiment somewhere.

The opsin now is being identified in the figure legend:

Fig. 5. Interluminescence is dependent on presynaptic vesicle release. **a** Schematics of synapses receiving synaptic blockers (SB) followed by electrical stimulation of presynaptic neurons together with CTZ application without (i) and after 48 h Botulinum Neurotoxin (BoNT) treatment (ii). **b** Representative trace of MEA recordings of **opsin (Chr2(C128S))** expressing neurons without BoNT treatment (i, from Fig. 4dii) and 48 h after BoNT treatment of the co-culture (ii): electrical stimulation of pre-synaptic neurons together with CTZ application fails to elicit firing after BoNT treatment, while blue light still induces firing in the same recording. **c** Ladder plots of recording conditions as in (b) from electrodes across populations (comparisons are: spontaneous activity vs SB addition (after), n=21, p<0.0001; electrical stimulation (after) vs immediately following CTZ (after), n=21, p=0.7173; electrical stimulation + CTZ (after) vs blue light (before), n=21, p=0.6055; blue light (before) vs blue light (during), n=21, p<0.0001; Wilcoxon matched-pairs signed rank test). The artifacts due to addition of reagents in MEAs are overlaid by a vertical white bar in the recording traces (the white gap right after addition of SB or CTZ). Artifacts due to electrical stimulation are visible under the red bolts. ns, not significant; ----, p<0.0001

6. Figure 7:

a. I like a; could be even more clear if you showed where you injected which virus in the anatomy cartoon.

b. Insets for b are difficult to read; move to separate panel if possible.

Figure 7 has been modified following the reviewer's suggestions.

7. Figure S1:

a. Please add time scale bar

Time scale bar has been added.

b. Not sure what 10 μ m means for CTZ...10 μ l? μ M?

This typo has been corrected.

8. Figure S3 & S4:

a. Please indicate which opsin is used for these experiments somewhere in figure or legend.

Added the information in both figure legends:

Supplementary Figure 3. Postsynaptic activity can be elicited with presynaptic electrical stimulation and with presynaptic bioluminescence through CTZ application. a Illustration showing luciferase expressing presynaptic cortical neuron (red) with its postsynaptic hippocampal neuron expressing the opsin ChR2(C128S) (green). b-e Illustrations with different treatment conditions (left panels) and corresponding representative recordings from the same co-culture (right panels) for the different treatments. b Spontaneous and synchronous firing activity in both pre- (marked by red arrow brackets) and postsynaptic neurons (marked by green arrow brackets). c Electrical stimulation of cortical presynaptic neurons (red boxed electrodes: upper half of MEA) evoked a strong time-locked increase in firing activity of post-synaptic neurons (lower half of MEA). Right (green boxed): zoom-in of one example postsynaptic electrode (vertical dashed line indicates the time of presynaptic electrical stimulation shown as red bolt). d Exposure of the culture to blue light (blue boxed electrodes) increases activity selectively in the opsin expressing post-synaptic population. Right (green boxed): zoom-in of one example postsynaptic electrode before and with blue light stimulation. e Exposure of the culture to CTZ (orange boxed electrodes) increases activity selectively within the opsin expressing post-synaptic population. Right: zoom-in of one example electrode each for the luciferase expressing pre-synaptic population (red boxed) and the opsin expressing postsynaptic population (green boxed). The artifacts due to addition of reagents in MEAs are overlaid by a vertical white bar in the zoomed-in MEA recording traces (the white gap right after addition of CTZ).

Supplementary Figure 4. Synaptic Blockers for isolating Interluminescence effects. a Representative example of an MEA recording from a cortical and striatal neuron co-culture with SB added about 1/3 into the recording (indicated by downwards black arrows), virtually silencing all activity in both cortical (red boxed) and striatal (green boxed) populations. b Zoomed-in representative traces of electrodes from striatal neurons demonstrating the silencing effect with addition of blockers (SB in I, from Fig. 4bI and II), while preserving the ability to elicit activity in opsin (ChR2(C128S)) expressing striatal neurons by blue light (II). c Ladder plots of recordings under the conditions depicted in (b), n=27, p<0.0001; Mann-Whitney test. The artifacts due to addition of reagents in MEAs are overlaid by a vertical white bar in the recording traces (the white gap right after addition of SB). ****, p<0.0001

Reviewer #3 (Remarks to the Author):

The authors have nicely addressed the reviewers' comments. I have just some minor concerns as follows.

1. It is not confirmed, how the close proximity of luciferases interrupts the synaptic cleft, however the expression of opsin closely affects total bioluminescence transmission. How much close proximity would influence the synaptic activation?

We appreciate the reviewer's thoughtful comment. We intend to address this and other questions about Interluminescence in much more detail than we were able to do here in upcoming studies.

2. The mixing of interluminescence and interluminescent seems not to sound for the general reader. They should carefully treat the word inappropriate expression. For example, the authors said, the precise photon density required to create the Interluminescent effects, however, this could be interluminescence effects.

This specific sentence has been changed:

hGtACR2 which have high photon sensitivity^{6,8,9,20}. The precise photon density required to create the ~~Interluminescent-Interluminescence~~ effects we observed is difficult to quantify in the abstract, as it depends on numerous factors including luciferase density, the impact of biologically specific

In addition, we consistently used the term Interluminescence throughout the manuscript except for the following two instances:

In our **Interluminescent** Optical Synapse, we used BL-OG to achieve synapse-specific and activity-dependent circuit control, by expressing the luciferase presynaptically and its partner

variables in the cleft (e.g., pH sensitivity) and specific details of synaptic connectivity (e.g., synaptic distance, number of **Interluminescent** synapses expressing both components, location of these synapses on the post-synaptic cell, etc.). The luciferase–opsin combinations used here

*3. In the following sentence, “the luciferin Coelenterazine (CTZ), the substrate for *Gaussia* luciferases”, there is a redundancy of luciferin and substrate. Therefore, they should use either luciferin or substrate.*

The sentence has been modified accordingly:

expressed. We then added the luciferin for *Gaussia* luciferases, Coelenterazine (CTZ), ~~the substrate for *Gaussia* luciferases~~, and observed increased (ChR2(C128S)) or decreased (hGtACR2) spontaneous activity consistent with the expressed opsin (**Fig. 2c**). By contrast, the

4. Why interluminescence doesn't have a volumetric effect than orthogonal neuropeptide? Is there any evidence of their claim?

In the Discussion we suggest that it is unlikely that luciferases outside the synaptic cleft have enough photon density to activate opsins outside of bona fide synapses. Different from neuropeptides diffusing away from a synapse, luciferases get dimmer over time and thus in a sense “self-inactivate”, making a volumetric effect very unlikely.

Synaptic vesicles are located in different subcellular domains of neurons including at presynaptic active zones, soma, dendrites, and axons. It is therefore possible and likely that luciferase is released at multiple sites following neuronal depolarization. Interluminescence likely reflects optical signaling at functional presynaptic synapses³⁹ because of the need for close proximity of luciferase and postsynaptic opsins across a shared synaptic cleft. Interluminescence outside of bona fide synapses is unlikely although this possibility could be explored in the future. Further, given that 20 seconds after release addition of luciferin does not elicit a postsynaptic response, it is most likely that luciferases diffused away from the synaptic space do not have a photon density high enough to activate opsins along the neuron. Thus, in contrast to neuropeptide transmission Interluminescence does not seem to have a volumetric effect.

5. In Figure 7, the inside letters are too small. They should enlarge the font sizes.

We moved the insets to separate displays with larger font.